# Early childhood education and care quality and associations with child outcomes: A meta-analysis

Antje von Suchodoletz[1,2]*, D. Susie Lee[3,4], Junita Henry [5], Supriya Tamang[6], Bharathy Premachandra[7], Hirokazu Yoshikawa[2,3]

1 Department of Psychology, New York University—Abu Dhabi Campus, Abu Dhabi, United Arab Emirates, 2 Global TIES for Children Research Center, New York University, New York, NY, United States of America, 3 Faculty of Arts and Science, New York University, New York, NY, United States of America, 4 Max Planck Institute for Demographic Research, Rostock, Germany, 5 Department of Global Health, Institute for Life Course Health Research, Stellenbosch University, Stellenbosch, South Africa, 6 Baltimore City Public Schools, Baltimore, MD, United States of America, 7 Department of Communication, Cornell University, Ithaca, NY, United States of America

* avs5@nyu.edu

**Data Availability Statement:** All relevant data are within the paper and its Supporting Information files.

## Abstract

### Objectives

The effectiveness of early childhood education and care (ECEC) programs for children's development in various domains is well documented. Adding to existing meta-analyses on associations between the quality of ECEC services and children's developmental outcomes, the present meta-analysis synthesizes the global literature on structural characteristics and indicators of process quality to test direct and moderated effects of ECEC quality on children's outcomes across a range of domains.

### Design

A systematic review of the literature published over a 10-year period, between January 2010 and June 2020 was conducted, using the databases *PsychInfo*, *Eric*, *EbscoHost*, and *Pubmed*. In addition, a call for unpublished research or research published in the grey literature was sent out through the authors' professional network. The search yielded 8,932 articles. After removing duplicates, 4,880 unique articles were identified. To select articles for inclusion, it was determined whether studies met eligibility criteria: (1) study assessed indicators of quality in center-based ECEC programs catering to children ages 0–6 years; and (2) study assessed child outcomes. Inclusion criteria were: (1) a copy of the full article was available in English; (2) article reported effect size measure of at least one quality indicator-child outcome association; and (3) measures of ECEC quality and child outcomes were collected within the same school year. A total of 1,044 effect sizes reported from 185 articles were included.

### Results

The averaged effects, pooled within each of the child outcomes suggest that higher levels of ECEC quality were significantly related to higher levels of academic outcomes (literacy, n =

**Funding:** Specific grant number: GST03 Initials of authors who received each award: AvS Full names of commercial companies that funded the study or authors: no commercial company funded the study or authors; the funding was provided by the Global TIES for Children Research Center at New York University Abu Dhabi URL to sponsor' website: https://nyuad.nyu.edu/en/research/faculty-labs-and-projects/global-ties-for-children.html The funders (other than the named authors) had no role in study design, data collection and analysis, decision to publish, or preparation of the manuscript.

**Competing interests:** The authors have declared that no competing interests exist.

99: 0.08, 95% C.I. 0.02, 0.13; math, n = 56: 0.07, 95% C.I. 0.03, 0.10), behavioral skills (n = 64: 0.12, 95% C.I. 0.07, 0.17), social competence (n = 58: 0.13, 95% C.I. 0.07, 0.19), and motor skills (n = 2: 0.09, 95% C.I. 0.04, 0.13), and lower levels of behavioral (n = 60: -0.12, 95% C.I. -0.19, -0.05) and social-emotional problems (n = 26: -0.09, 95% C.I. -0.15, -0.03). When a global assessment of child outcomes was reported, the association with ECEC quality was not significant (n = 13: 0.02, 95% C.I. -0.07, 0.11). Overall, effect sizes were small. When structural and process quality indicators were tested separately, structural characteristics alone did not significantly relate to child outcomes whereas associations between process quality indicators and most child outcomes were significant, albeit small. A comparison of the indicators, however, did not yield significant differences in effect sizes for most child outcomes. Results did not provide evidence for moderated associations. We also did not find evidence that ECEC quality-child outcome associations differed by ethnic minority or socioeconomic family background.

## Conclusions

Despite the attempt to provide a synthesis of the global literature on ECEC quality-child outcome associations, the majority of studies included samples from the U.S. In addition, studies with large samples were also predominately from the U.S. Together, the results might have been biased towards patterns prevalent in the U.S. that might not apply to other, non-U.S. ECEC contexts. The findings align with previous meta-analyses, suggesting that ECEC quality plays an important role for children's development during the early childhood years. Implications for research and ECEC policy are discussed.

## Introduction

Over the past decades, the number of children participating in early childhood education and care (ECEC) programs increased worldwide [1]. Children's experiences in ECEC programs have thus become an important factor in their development during the early childhood years [2, 3]. The expansion of ECEC provision worldwide is in line with global policy efforts, such as the Sustainable Development Goal (SDG) goal 4 and specifically target 4.2, which calls for universal access to one year of preprimary education [4, 5]. ECEC programs are commonly implemented with the goal to "enhance early learning and development" and/or "increase opportunities for all children to succeed in school" [6, p. 3]. Many calls have been made for high-quality programs, including in the wording of SDG 4 itself ("ensure that all girls and boys have access to quality early childhood development, care and pre-primary education" [5]. Moreover, the G20 Initiative for Early Childhood Development [7] emphasizes the importance of political buy-in and state and non-state investments in the early years in order to narrow achievement and opportunity gaps that exist between children from higher and lower socioeconomic backgrounds. The effectiveness of ECEC programs for children's development in various domains has been well documented, both short-term and long-term, although short-term effects have been larger [8, 9]. Yet, the evidence base on the associations of quality of ECEC with children's developmental outcomes is lacking accuracy in at least three ways. First, the literature is dominated by research from countries with a long history of ECEC provision, such as the United States (U.S.) and European countries like Finland, Germany, the Netherlands, or the United Kingdom, resulting in a scarcity of systematic research syntheses that

integrate the global literature on ECEC to build knowledge about children's educational opportunities across the world. Second, a large body of original research and research syntheses investigates the role of structural aspects of the ECEC program. However, a pending question remains whether such structural characteristics itself systematically change the effects of process quality on child outcomes, or whether process quality changes the impact of structural characteristics on child outcomes. And third, there is a need to broaden the scope of developmental outcomes. Participation in ECEC programs may also have impacts on other, non-academic outcomes, such as social-emotional skills that have become a major focus of ECEC programs and interventions [10]. To address these gaps, our primary goal in the current meta-analysis was to provide a synthesis of available international evidence to better understand the mechanisms underlying the pathways from ECEC quality to child outcomes.

## Defining ECEC quality: What matters for children's outcomes?

Although there is a growing consensus that the level of quality of ECEC services influences children's developmental outcomes [3, 11], the definition of ECEC quality has been a topic of debate since the 1970s, a debate that has undergone continuous change paralleling changes in socio-political structures and influences [2]. More recent definitions refer to ECEC quality as a multidimensional construct that includes structures of, and processes and practices in ECEC settings [12]. This is reflected in ECEC policy with quality standards being used as regulations with "the assumption that improving structural or process quality improves children's outcomes indirectly or directly" [3, 6 p. 4]. Yet, the empirical evidence relating ECEC quality indicators with child outcomes is mixed and inconsistent in size [3], raising conceptual and methodological concerns regarding ECEC quality models [6].

**Structures and processes.** Structural aspects are major factors of ECEC programs [13, 14]. The most commonly studied structural aspects are teacher-child ratio and class size [3]. Evidence shows moderate positive effect sizes of fewer children per teacher and smaller class sizes on children's outcomes [13, 15]. Yet, not all studies have found significant associations. For example, a meta-analysis from the U.S. found teacher-child ratio and class size to be unrelated with children's language, reading, math and social skills [16]. It may be that, as regulation advances in particular countries, variation in these structural dimensions decreases and therefore the predictive power of ratios and class sizes is limited. A second major area of structural quality includes teacher factors, such as training, education, and experience. Associations between these factors and children's outcomes are also inconsistent. The above cited meta-analysis found significant positive, yet small effects of teacher factors on children's language, reading and math skills [16]. Other studies, however, did not find significant associations [17–19]. For example, data from a U.S. state's Quality Rating and Improvement System showed that neither general teacher education (i.e., type of degree) nor additional ECEC training were associated with children's school readiness skills [18]. Although one might conclude from the inconsistent empirical evidence for many structural aspects that these indicators of ECEC quality do not matter for children's outcomes to the anticipated extent, it remains important "to continue examining pathways from structural quality to children's outcomes to inform policy and practice" [6 p. 7, 16]. Because structural features of ECEC settings have been more regulable, many countries focus on structural standards as key strategy for improving the quality of ECEC programs [3, 17]. The G20 Development Working Group [7] urges to focus on the quality of the infrastructure and capacity building of, decent work conditions for and adequate training of the ECEC workforce. Government regulations can set standards for these features, for example, by lifting the minimum requirements for teacher-child ratio or requiring a certain percentage of teaching staff to be qualified in early childhood education. Such

structural regulations determine the setting in which children learn and thus may be important preconditions for process quality [3].

Over the past decade, aspects of process quality—teacher-child interactions and the level of stimulation of early learning in particular—have become an important focus of the effort to raise ECEC quality [3]. Socioecological, attachment, and learning theories recognize that teacher-child interactions provide an important context for children's development and learning [6, 20]. Research, predominantly from the U.S., indicates small but statistically significant positive associations between the quality of teacher-child interactions and children's academic and social-emotional outcomes [21–23]. Associations for academic outcomes are often stronger than for socio-emotional outcomes [24]. However, depending on the domain of teacher-child interactions assessed in a study, estimated effects vary in size with some research also finding no significant associations. Theoretically it is thought that frequent warm and emotionally supportive interactions between teachers and children foster gains in children's skills. Yet, effect sizes for teacher provision of emotional support are often small (0.01–0.08) when controlling for earlier child skills [13, 25, 26]. Teacher-child interactions characterized by conflict, tension and anger, in contrast, have been linked with a higher chance that children develop achievement or behavioral problems [27–29]. A recent *Starting Strong* report [3] found a significant negative summary effect size of negative teacher-child interactions across multiple studies (-0.33). Several studies documented the benefits of well-organized classrooms for children's development, both academically [30] and behaviorally [31]. Leyva et al. [26] conclude that effect sizes for this aspect of teacher-child interactions were higher, ranging between 0.14–0.49 (p. 784). Finally, clear instruction intended to enhance knowledge of concepts and language, tying new facts to children's prior knowledge, and providing immediate, specific feedback [20], predicts growth in children's academic skills [13, 25]. Effect sizes have been reported to be small to moderate (0.002–0.32, cited from [26, p. 784]. However, as with other aspects of teacher-child interactions, not all studies reported significant associations between the level of instruction provided by the teacher and children's academic skills [24, 32].

Different approaches to the measurement of process quality might have contributed to the inconsistent empirical evidence for the link between process quality indicators and child outcomes. The two most common approaches to the measurement of process quality are observations and teacher self-reports. The Classroom Assessment Scoring System (CLASS) [33] and the Early Childhood Environment Rating Scale (ECERS) [34], for example, are often used as observational tools to assess indicators of process quality in ECEC programs in the U.S. and internationally. The most widely used self-report questionnaire is the Student-Teacher Relationship Scale (STRS) [35] to assess "the teacher's perception of, and feeling about, the child's behavior toward her" which are thought to be a key component influencing the teacher's ability to engage in positive interactions with the child [36, p. 126]. Compared to observational measures, the use of self-reports is often more economic. However, self-report data can be influenced by self-representation bias or social desirability bias, "a tendency of individuals to present themselves and their practices in a favorable way" [37, p. 628]. Thus, it is important to test whether observational and self-report measures yield the same results. Comparability across studies is also often challenged by differences in the data collection schedule (beginning, mid, or end of school year; cross-sectional or longitudinal).

**ECEC quality indicators as moderators.**   In the past, most research has tested direct associations between ECEC quality indicators and children's skills and knowledge. As reviewed above (and in other sources, e.g., 6), the strength of direct associations is small to moderate, yet, with a considerable number of studies finding no significant links. Indirect associations have been examined to a lesser extent despite the fact that predominant conceptual models assume process quality as a mechanism underlying the association between structural

characteristics and child outcomes [6, p. 6]. The NICHD Early Child Care Research Network [38] study provided initial evidence for indirect relations from structural aspects of ECEC program to child outcomes (cognitive and social competence) through the quality of processes in the ECEC program. However, the indirect associations were very small and findings have not been replicated in some other large-scale data sets [6].

A better understanding of the underlying processes linking ECEC quality with child outcomes may be gained by testing interaction effects of ECEC quality indicators. It is possible that it is a specific combination of structural and process aspects that matters for children's outcomes. For example, it has been found that associations between process quality and children's social-emotional skills were moderated by dosage. Children who spent more time in high-quality ECEC settings were reported to have higher levels of social-emotional skills compared to children who spent less time in high-quality ECEC settings [39, 40]. Such results suggest that structural characteristics can reinforce positive effects of high levels of process quality as well as negative effects of low levels of process quality. Likewise, positive effects of the level of instructional processes on children's gains in literacy and numeracy might only be present in small classes where teachers can engage in differentiated instruction, whereas in large classes such an effect might be absent. However, results are mixed and other studies did not find significant results when testing structural characteristics of the ECEC setting as moderators of the association between process quality and child outcomes [41].

Alternatively, it might also be possible that associations between structural aspects and child outcomes will be stronger under high levels of process quality, compared to low levels of process quality. For example, a study found that teacher emotional support moderated the association between classroom composition (i.e., high levels of problem behaviors in the classroom) and children's relational functioning. The negative effect of a highly challenging class on individual children's relational functioning was buffered by teachers who were highly emotionally supportive [42]. Although fewer studies tested the moderating role of process quality, it can provide important information about the mechanisms underlying the associations between ECEC quality indicators and child outcomes.

## Different family and economic backgrounds

The policy focus on ECEC quality is driven by the assumption that participation in ECEC can compensate for educational disadvantages associated with low family socio-economic status and ethnic minority status. Indeed, evaluations of ECEC programs (for example, Head Start in the U.S.) provide evidence for this assumption, suggesting that program effects may be largest for children from disadvantaged backgrounds [43–46]. ECEC programs have the potential to compensate for educational disadvantages by providing rich and engaging learning environments and to support these children to catch up with their peers [47]. As such, ECEC programs can disrupt trends leading to achievement gaps which have been found to start prior to age three [44]. Yet, to date, systems, including ECEC, continue to perpetuate racism and inequities, thus "reduc[ing] opportunities for certain groups to thrive and meet their potential" [44, p. 65]. In order to strengthen the impact of early learning, more effective, evidence-based policies are thus needed.

**Children from ethnic minority backgrounds.** Children from ethnic minority backgrounds, many of whom grow up in bilingual environments with different languages spoken in the home and at school [48], often experience difficulties at school which can lead to school readiness and achievement gaps [49, 50]. Several explanations are discussed in the literature, such as racism and lack of attention to culture [51], differences in parental involvement in children's education [52], in child-rearing and education-related beliefs and practices [53], in

parents' education [48], and in opportunities for informal learning at home [49]. In addition, differences in the quality of teacher-child interactions may contribute to the different school experiences of children from ethnic minority backgrounds. Teacher-child interactions operate similarly across ethnic groups [26, 54, 55]. Yet, the quality of instruction has been shown to decrease as the proportion of ethnic minority children in a classroom increases [56–58]. Such findings are of concern as they highlight the risk of ECEC programs not meeting the needs of ethnic minority children.

**Children from low-income families.**    High-quality ECEC programs are thought to promote equality in educational opportunities for all children, but especially for children from low-income families [2, 4–6]. There is consensus that these children, compared to their peers from more advantaged backgrounds, have a greater risk of school failure and adjustment problems that may start as early as preprimary age [3, 8, 47, 59]. Due to limited material resources or educational opportunities at home, these children may lack the skills necessary to succeed in school [59]. Large-scale interventions have been implemented to reduce socioeconomic achievement gaps. While beneficial effects have been demonstrated during the preschool years (such as cognitive gain and reduction of behavior problems), they often do not persist into kindergarten and school [8, 60]. Such results give reason to question whether ECEC services are reaching their potential, thus questioning whether these programs have the anticipated beneficial effects for children from low-income families [6].

## The present study

The present meta-analysis aims to contribute synthesized evidence from across the world on the developmental impact of ECEC to inform the formulation of an extended ECEC quality model for use in research, policy-making and practice. It also aims to address various shortcomings of prior meta-analyses, for example, the limitation to one geographical region [9, 19, 24], or in the child outcomes assessed (e.g., cognitive and academic outcomes) [19]. Specifically, we examined the magnitude of associations between ECEC quality indicators and a variety of child outcomes using studies from around the world. In doing so, we explored characteristics and mechanisms that may underlie the associations between ECEC quality and child outcomes. The additional questions are: (a) Do ECEC quality-child outcome associations differ by quality indicator (structural versus process) or by different aspects of process quality (emotional support, instructional support, classroom management)? (b) To what degree does one indicator of ECEC quality (structural or process) moderate the associations between the other indicator of ECEC quality (structural or process) and child outcomes? (c) Do process quality-child outcome associations differ by the method of process quality assessment (observed versus teacher-report)? (d) Do ECEC quality-outcome associations differ by timing (beginning versus end of school year; concurrently versus longitudinally)? And (e) Do ECEC quality-child outcome associations differ by family ethnic minority status and socio-economic background?

## Method

### Literature search, inclusion criteria, and coding

**Search procedures.**    We conducted a systematic review (the review was not registered) of the global literature published over a 10-year period, between January 2010 and June 2020, using the databases *PsychInfo*, *Eric*, *EbscoHost*, and *Pubmed*. We based these dates on previous meta-analyses investigating associations between various aspects of ECEC quality and child outcomes for which literature searches included articles published prior to 2010 [e.g., 24, 61–67]. We allowed overlap in years with the previous meta-analyses and stopped in 2020, at a time when ECEC provision was challenged because of the COVID-19 pandemic. The time

period (2010–2020) also covers a period of ECEC-focused policy initiatives across the world, for example, the national plan for medium and long-term education reform and development (2010–2020) in China [68], recommendations on high-quality ECEC systems by the [69], or the G20 Initiative for Early Childhood Development [7] and G20 Education Ministers' Declaration [70].

Searches focused on early childhood education; additional search strings included keywords for ECEC quality indicators (process quality and structural characteristics) and child outcomes (for a complete list of keywords see S1 File). In addition, a call for unpublished research or research published in the grey literature was sent out through the authors' professional network to reduce publication bias. In total, 170 scholars in research institutions, NGOs, and public institutions worldwide were contacted via email and asked to share their work on this topic. This call yielded 40 studies. Prior to the coding process we checked if any of these studies were published in the meantime and used the published version of the article. This was the case for two studies (published in 2021). Together, this search yielded 8,932 articles. After removing duplicates, 4,880 unique articles were identified.

**Inclusion criteria.** To select articles for inclusion in the meta-analysis, it was first determined whether studies met eligibility criteria: (1) the study assessed indicators of quality in center-based ECEC programs catering to children ages 0–6 years; and (2) the study assessed child outcomes. This was done by reading titles and abstracts. To ensure intercoder agreement, 20% of studies were screened by two independent coders. The agreement between coders was, on average, 86%. During this phase, 808 articles met the eligibility criteria. Second, the articles were screened for three additional inclusion criteria: (1) a copy of the full articles was available in English; (2) the article reported effect size measure of at least one quality indicator-child outcome association; and (3) measures of ECEC quality and child outcomes were collected within the same school year. During this step, meta-analyses, literature reviews and systematic syntheses of multiple datasets were excluded. Longitudinal studies were only excluded when child outcome measures were from a different school year than ECEC quality measures. Intervention studies were excluded unless relevant effect size measures were reported prior to the intervention. Intercoder agreement was, on average, 83% for 20% of the studies. After excluding articles that did not meet the inclusion criteria, 265 articles were eligible for coding. During coding, 77 articles were excluded. Reasons for exclusion during the coding included the following: that the study did not differentiate between different school years in the analysis ($n = 43$); did not report relevant effect sizes ($n = 11$); used child outcome as predictor of ECEC quality ($n = 3$); or the ECEC quality measure was collected after child outcome was assessed ($n = 9$). The selection process is shown in a PRISMA (Preferred Reporting Items for Systematic Reviews and Meta-Analyses) flowchart in Fig 1. An overview of all included studies is shown in S1 Table of S2 File.

**Coding of study characteristics.** Copies of eligible articles were obtained. Articles were coded by two coders. Coding accuracy was ensured by double-coding 20% of articles by both coders; intercoder agreement was high (91%). Disagreements between coders were discussed; the consensus became the code. We recorded several key study features: year of publication, country where data was collected, name of the study from which data was reported, number of participants (teacher, child), where participants were recruited (ECEC setting or through other sources), age and gender of participants, percentage of children in the sample from low-income background and from ethnic minority background, and the time in the school year when the ECEC quality and child outcome measures were collected (beginning/mid/end of school year).

We also recorded structural and process indicators of ECEC quality, and associations between these indicators with a child outcome reported in each study. Five structural indicators were coded, differentiating between classroom-level (two indicators: teacher-child ratio,

| Systematic Literature Search (January 2010-June 2020) | | | | | | | | | |
|---|---|---|---|---|---|---|---|---|---|
| Hits | | | | | | | | | 11850 |
| Exported | | | | | | | | | **8892** |
| | | *PsychInfo* | | *ERIC* | | *EcbsoHost* | | *PubMed* | |
| | | Hits | Exported | Hits | Exported | Hits | Exported | Hits | Exported |
| Titles | | 28 | 28 | 28 | 28 | 206 | 92 | 8 | 8 |
| Keywords | | 180 | 180 | 0 | 0 | 1759 | 1580 | 1045 | 1045 |
| Abstract | | 1380 | 1378 | 896 | 890 | 6049 | 3354 | 309 | 309 |
| n= | | 1588 | 1586 | 924 | 918 | 8014 | 5026 | 1362 | 1362 |

| Total number of articles identified | |
|---|---|
| Systematic Literature Search | 8892 |
| Email Call - Gray Literature: Published | 5 |
| Email Call - Gray Literature: Unpublished | 17 |
| Email Call - Other | 18 |
| **Total** | **8932** |

Duplicates removed (N=4052)

Total number of unique identified articles (N= 4880)

| Prescreened articles | | | Further duplicates identified during prescreening process | |
|---|---|---|---|---|
| 529 | US based studies with quality and child outcome measures | | 6 | US based studies with quality and child outcome measures |
| 294 | Non-US based studies with quality and child outcome measures | | 9 | Non-US based studies with quality and child outcome measures |
| 3893 | Studies with no ECE quality and child outcome measures | | 45 | Studies with no ECE quality and child outcome measures |
| 164 | Studies not online or studies in another language | | 4 | Studies not online or studies in another language |
| **4880** | **Total** | | **64** | **Total** |

Total number of articles eligible for screening: N=808

| Screening for eligibility to code (full text) | | | Further duplicates identified during screening process | |
|---|---|---|---|---|
| 498 | Do not code | | 3 | Do not code |
| 6 | Non-English publication (exclude) | | 0 | Non-English publication (exclude) |
| 34 | Meta analysis, literature review or synthesis (exclude) | | 1 | Meta-analysis, literature review or synthesis (exclude) |
| 270 | Code: Meets all eligibility criteria | | 5 | Code: Meets all eligibility criteria |
| **808** | **Total** | | **9** | **Total** |

Total number of articles eligible for coding: N=265

| Number of articles excluded during coding process | | | Further articles excluded during the data preparation phase | |
|---|---|---|---|---|
| 43 | Study does not differentiate between different school years in the analysis | | 2 | Reported effect sizes could not be standardized because the required standard deviation information was not reported |
| 10 | Study does not report relevant effect sizes | | 1 | Effect sizes were insignificant and authors did not report exact estimates |
| 10 | Quality measures collected after child outcome | | | |
| 6 | Intervention study that does not include pre-test associations | | | |
| 4 | Study does not meet process quality or child outcome criteria | | | |
| 3 | Study uses child outcomes as a predictor | | | |
| 1 | Duplicate detected during coding | | | |
| **77** | **Total** | | **3** | **Total** |

Total number of articles included in the meta-analysis: N=185

| Additional details | |
|---|---|
| 13 | Averaged process quality across two timepoints, in this case reported as a second timepoint |
| 6 | Measured child outcomes at the class level |
| 2 | Insignificant effects not reported by author, exact estimates reported in this study as missing and insignificant |
| 13 | Time of the year during which measures were collected are missing |

**Fig 1. PRISMA flowchart of article selection.**

group size) and teacher-level factors (three indicators: teacher age, teacher education [coded as percentage of teachers in the sample with a multi-year college or university degree], years of experience). Four process quality indicators focused on the quality of interactions and instruction: emotional quality (emotional support/closeness/positive climate/responsiveness), instructional quality (stimulation of cognitive and language development), managerial quality (classroom organization/classroom management/chaos [reversed coded–absence of chaos]), and conflict/negative climate [reversed coded–absence of conflict]. To categorize items or

scales of measures into these four indicators, we relied on the description of the measure and labeling by the author(s) of the original study. If the study used a global process-quality score (i.e., the score did not differentiate between specific indicators but instead reflected an average level of process quality across multiple indicators) this was recorded as a separate process quality indicator. When multiple measures within one domain were reported, the average was calculated and used in further analyses.

We recorded any association measures reported between ECEC quality and eight types of child outcomes. Types of child outcome included academic outcomes (separately for math and literacy/language), behavioral skills (such as self-regulation, executive function, positive learning behaviors), social competence, behavioral problems (such as aggressive behavior, conduct problems), social-emotional problems (such as withdrawal, anxious/depressed), and motor skills. If the study used a global score of child outcome, this was recorded. When multiple outcomes within one category were reported, the average was calculated and used in further analyses.

In addition, we coded whether an observational or self-report (i.e., teacher report) measure was used for the particular aspect of process quality. Finally, we recorded whether zero-order correlation (coefficient without covariates) or regression coefficient (coefficient with covariates), or both, were reported for an association between process quality and child outcome. If both types of coefficients were provided for the same process quality-child outcome association, both coefficients were coded.

The coding was done using Excel where the coding sections were detailed. An amendment was made to include process quality descriptive information (Mean, SD) as an additional Excel document. Both documents are available in the Data and Results of S13 File.

**Data analytic plan.**   Our measure of effect size to examine the associations between ECEC quality indicators and child outcome domains was the strength and direction of association, or correlation, between an ECEC quality measure and a child outcome measure (the document is available in the Data and Results of S13 File). Most studies reported correlation indices in the form of either zero-order correlation coefficient or regression coefficient that were both standardized (by multiplying the unstandardized coefficient by the coefficient of the standard deviation of an ECEC quality measure divided by the standard deviation of a child outcome measure), such that the range of a coefficient is between -1 to 1. Three studies were excluded from the meta-analysis during the data preparation phase. The effect sizes reported in two studies could not be standardized because the required standard deviation information was not reported; the other study reported insignificant findings but did not include the exact estimates which were recorded as missing during the coding (Fig 1). Out of the remaining 1,112 unique effect sizes that were recorded, 68 were excluded because the standardized effect size values did not range between -1 and 1. As a result, a total of 1,044 effect sizes reported from 185 articles were eligible for meta-analysis. These standardized effect sizes were then transformed into Fisher's $z$-scale, to normalize the sampling distribution of effect sizes, using the 'escalc' function from the R package 'metafor' [71].

For all meta-analyses, two sets of models were estimated–one with and one without control variables regarding child sample characteristics: sex composition (proportion of girls in the sample) and average age of children in the sample. These two variables were standardized before entering analyses. However, out of the 185 studies, 23 studies did not have either of these variables available. To maximize the number of studies available to estimate pooled effect sizes (overall association between ECEC quality indicators and child outcomes), we first estimated a model based on all available 185 studies, and then estimated another model based on a subset of studies that reported the control variables, so that a model with and without these variables could be compared.

For the meta-analytic models estimated, we report tau-squared ($\tau^2$) which captures the absolute measure of between-study variance in random-effects meta-analysis. We also report I-squared ($I^2$) which describes the percentage of variation across studies that is due to heterogeneity rather than chance, and is calculated as the ratio of true heterogeneity to total variance across the observed effect sizes [72]. Rho, a within-study effect size correlation, was set at 0.8 for the analyses, and a sensitivity analysis was conducted to check if $\tau^2$ and average effect size estimates were robust to different values of rho ranging from 0 to 1.

To examine the overall magnitude of associations between ECEC quality indicators and child outcomes, we averaged effects pooled within each of the child outcomes via eight random-effects models. We used robust variance estimation to calculate the pooled effect sizes in which weights of each estimate of association were based on the working models that effect sizes are correlated within studies. Details on the formulas specifying the correlated effects covariance structure and weights calculation can be found in [73, p. 4]. Structural and process indicators of ECEC quality were combined to maximize the number of included effect sizes. Because effect sizes for each child outcome could include the association between several quality indicators and the child outcome category, some studies contributed multiple effect sizes for each analysis. To handle within-study dependence among effect sizes, robust variance estimation was implemented for all meta-analyses conducted [74, 75], using the 'robumeta' package in R [76]. This approach followed previous studies that dealt with the similar issue of correlated effect sizes within studies [77, 78].

In addition, we explored characteristics and mechanisms that may underlie the associations between ECEC quality and child outcomes. We describe the analytical approach below for each additional question:

a. *Do ECEC quality-outcome associations differ by quality indicators (structural versus process) or by different domains of process quality*? We first tested the associations between structural characteristics and process quality indicators with child outcomes separately, using the same analytical approach as described above. Next, we conducted moderator analyses using mixed-effects meta-regression models. Specifically, a binary variable of whether an effect size was for a process quality indicator-child outcome association or a structural indicator-child outcome association was added to the models described above. As such, effect sizes that involved a structural indicator of ECEC quality were compared with those that involved a process indicator of ECEC quality, regardless of whether an effect size came from the same study or not. Mixed-effects model allowed for adjusting within-study clustering of effect sizes. We further examined if the associations differed by three domains of process quality: instructional quality, emotional quality (reflects the presence of emotional support and absence of conflict), and managerial (reflects the presence of classroom management and absence of chaos). To test this difference, we estimated the same models as above, but restricted the analysis to process quality-child outcome associations and added a categorical variable indicating whether an effect size was for one of the three domains of process quality. Due to too few studies available (less than 3), we could not reliably estimate models involving motor skills and thus do not present results on motor skills.

b. *To what degree does one indicator of ECEC quality (structural or process) moderate the ECEC quality-child outcome associations of the other indicator of ECEC quality (structural or process)*? We first tested whether structural indicators moderated process quality-child outcome associations. For that, we restricted data to effect sizes for process quality-child outcome associations and to studies that reported measures for at least one of the five structural indicators (teacher-child ratio, group size, teacher age, teacher education, teacher's years of teaching in ECEC settings). Because not all studies reported the five structural

indicators simultaneously, separate mixed-effects models were estimated for each structural indicator within each child outcome, to test if process quality-child outcome associations differed by the degree of a structural indicator. To do so, we added a structural indicator variable to a model, where the outcome variable was an effect size for process quality-child outcome associations. We followed the same approach for testing whether process quality moderated structural indicator-child outcome associations. For these analyses, the data were restricted to effect sizes for structural indicator-child outcome associations and to studies that reported mean scores of measures for at least one of the five process quality indicators (emotional quality, instructional quality, managerial, conflict, and global score). The available effect sizes were grouped by child outcome to test whether structural indicator-child outcome associations differed by the degree of a process quality indicator. A process quality indicator variable was added to a model where the outcome variable was an effect size for structural indicator-child outcome associations. For both sets of analyses, we could not reliably estimate models involving motor skills because models did not converge due to the small number of studies available.

c. *Do process quality-child outcome associations differ by the method of process quality assessment (observed versus teacher-report)*? Here the goal was to compare effect sizes reported from the same studies rather than estimating pooled effect sizes across studies. We prepared a dataset with those studies for which the same process quality-child outcome association was reported twice, once using an observed process quality measure and once using a self-reported process quality measure. To compare effect sizes, we used Kruskal-Wallis non-parametric test because the normality assumption was not met for the effect size values.

d. *Do process quality-child outcome associations differ by timing (beginning versus end of school year; concurrently versus longitudinally)*? The dataset for this analysis included only studies that reported the same type of quality-child outcome association at different time points during the school year (beginning versus end of school year; at the same versus at different time points during the school year). To compare effect sizes, we used Kruskal-Wallis non-parametric test because the normality assumption was not met for the effect size values.

e. *Do ECEC quality-child outcome associations differ by family ethnic minority and socio-economic background*? We built on the models for estimating the overall pooled effect sizes but instead of random-effects models, mixed-effects models were used by simultaneously estimating fixed-effects of the percentage of children in a study sample from family ethnic minority background or low-income background. In these models, the percentage of children in a study sample from a minority or a low-income background was used. The coefficients for the background moderation effect were multiplied by 10 to aid interpretation (i.e., 1 unit equals 10% change in the proportion of children from minority/low-income background). The models were restricted to studies from high-income countries as this information was not reported in studies from low-to-middle income countries. Effect sizes involving motor skills did not enter the analysis, because there was only one study that reported both information.

**Sensitivity analyses.** We undertook different sets of sensitivity analyses, mainly for the pooled effect sizes within each of the eight child outcome categories. First, we examined possible publication bias using significance funnel plots [79] (S3 File). Significance funnel plots display the same data (i.e., point estimates on x-axis and estimated standard errors on y-axis) as classic funnel plots; however, while the latter assumes that publication bias is less severe among large-sample studies, significance funnel plots help to assess publication bias based on

the assumption that publication bias operates on smaller *p*-values rather than sample size per se. As such publication bias based on selection (e.g., the file drawer problem) is well known in psychology, we chose to examine significance funnel plots assuming that selection acts at the alpha level of 0.05. Second, we examined whether pooled effect sizes changed after adding publication year as a covariate to the models estimated. A third sensitivity analyses examined whether pooled effect sizes changed if positive behavioral and social-emotional outcomes were grouped together. This approach was repeated with negative behavioral and social-emotional outcomes being grouped together. A final sensitivity analysis examined whether effect sizes differed depending on the type of effect size (effect size measure calculated with vs without covariates). This analysis used only studies that reported the same type of quality-child outcome associations assessed at the same time point using the same process quality observation method, so that any difference between effect sizes was attributable to the type of effect sizes. We again used Kruskal-Wallis non-parametric test to compare effect sizes reported from different conditions within studies (regression coefficient vs. zero-order coefficient) because the normality assumption was not met for the effect size values (S4 File).

## Results

### Descriptive overview

The meta-analysis included effect sizes from 185 studies with a total of 38,168 teachers and 229,697 children, from over 8,237 ECEC sites (information about number of sites was missing for 87 studies). Studies varied in sample size (teachers: 4–7,600, *M* = 240, *SD* = 743; children: 47–16,356, *M* = 1,248, *SD* = 2,317). Children were, on average, 54.41 months old (ranging from 14 to 75 months, *SD* = 10.72) and gender was similarly distributed (on average, 49% girls). On average, 66% of children in study samples were from low-income backgrounds and 58% from ethnic minority backgrounds. When reported, the majority of teachers were female (97%) and had 12 years of teaching experience (*SD* = 4.33). Information regarding class size was reported in 58 studies, with an average of 19 children per classroom (*SD* = 5.62, ranging from 7 to 35 children). Participants (teachers and children) were predominantly recruited in the ECEC setting.

The majority of studies (123 out of 185) reported research from the US. A breakdown of studies by United Nations Regional Groups (https://en.wikipedia.org/wiki/United_Nations_Regional_Groups) revealed that 39 studies were from countries grouped in the Western European and Others Group. Studies were also grouped by the World Bank classification of countries by income level (https://datahelpdesk.worldbank.org/knowledgebase/articles/906519-world-bank-country-and-lending-groups). According to this breakdown, 165 studies reported research from countries grouped high income (Table 1 for a summary). S1 Table in the S2 File shows an overview of the studies entered in the meta-analysis, including study and sample information, indicators of ECEC quality (structural and process), and child outcomes.

### What was the association between ECEC quality and child outcomes?

We first sought to establish the magnitude of the association between indicators of ECEC quality and child outcomes. We combined effect sizes for structural and process quality indicators; effect sizes were pooled with each of the 8 child outcome categories. Fig 2 summarizes the pooled effect sizes for the eight child outcome categories, based on the intercept-only models before adjusting for potential differences by child sex and age composition of the samples. We interpreted the pooled effect sizes as significant when the 95% Confidence Interval did not include zero.

Most child outcome categories showed a significant overall association with ECEC quality, with small effect sizes (in the description of results below, n refers to the number of unique

**Table 1. Number of studies providing usable effect size for the meta-analysis (total of 185).**

| Category | | # of studies |
|---|---|---|
| U.S. and others | U.S. | 123 |
| | Others | 62 |
| By region[1a1] | Africa | 3 |
| | Asia and the Pacific | 12 |
| | Eastern Europe | 0 |
| | Latin America and Caribbean | 8 |
| | Western Europe | 39 |
| By income[2] | Low to middle income | 20 |
| | High income | 165 |

[1] United Nations Regional Groups (link).

[2] World Bank Country and Lending Groups (link).

studies). Higher levels of ECEC quality were significantly related to higher levels of academic outcomes (literacy, n = 99: 0.08, 95% C.I. 0.02–0.13; math, n = 57: 0.07, 95% C.I. 0.03–0.10), behavioral skills (n = 65: 0.12, 95% C.I. 0.07–0.17), and social competence (n = 61: 0.13, 95% C.I. 0.07–0.19), and lower levels of behavioral (n = 61: -0.12, 95% C.I. -0.19 - -0.05) and social-emotional problems (n = 27: -0.09, 95% C.I. -0.15 - -0.03). For motor skills, however, the small number of studies and available effect sizes limited our ability to assess how robust the positive association between ECEC quality and motor skills would be (n = 3: 0.12, 95% C.I. -0.04–0.29). Of note, when a global assessment of child outcomes was reported, the association with ECEC quality was not significant (n = 14: 0.02, 95% C.I. -0.07–0.11). In addition, the 95% confidence

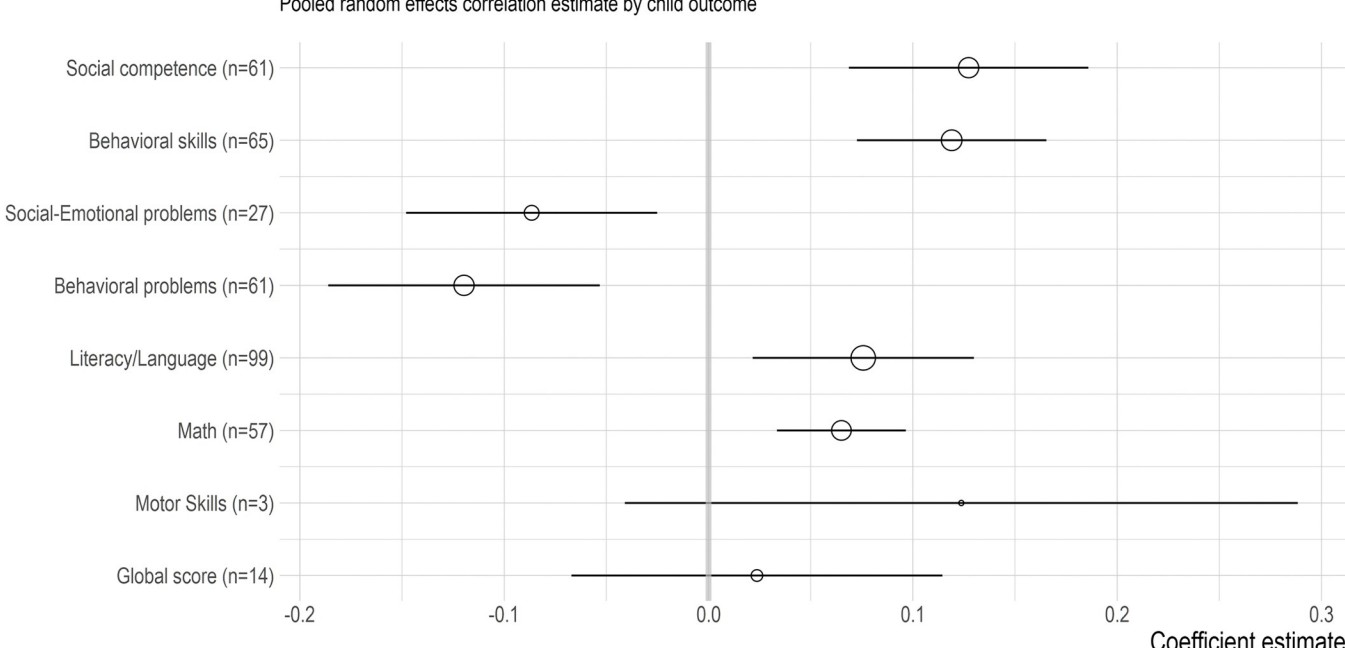

Pooled random effects correlation estimate by child outcome

**Fig 2. Pooled effect size estimates for ECEC quality-child outcome associations.** Pooled estimate for each child outcome category is shown in circles, which differ in size by the number of unique studies available (larger circles reflect a higher number of unique studies available; number of unique studies is given in the parentheses). The estimates' 95% confidence intervals are shown in black lines.

intervals suggested some uncertainty around the estimates. We repeated the analysis using only studies from high-income countries. The results replicated and are reported S5 File. However, for low-to-middle income countries, the same separate analysis could not be completed because of the small number of unique studies.

When significance funnel plots were examined (S3 File), there was evidence that the summary estimates were robust to publication bias for academic skills (math and language/literacy), behavioral skills, and social competence, suggested by the close distance between the pooled estimates across effect sizes (the black diamond) and within only the non-affirmative (i.e., *p*-value larger than 0.05) effect sizes (the grey diamond). In contrast, some evidence of publication bias was suggested for behavioral problems, social-emotional problems, and motor skills. Except for behavioral problems and social-emotional problems, publication bias was assumed to operate in a positive direction at an alpha level of 0.05.

**Control variables.** The sex composition (proportion of girls in the sample) was not associated with effect size magnitude for any the models. The average age of children in the sample (in months) was associated with effect size magnitude, however, only for the models that included behavioral problems and the global measure. Specifically, for the association between ECEC quality and behavioral problems (available studies: *n* = 51), 1 standardized unit (11.3) increase in the average age of children was associated with -0.07 (95% C.I. -0.13, -0.02) smaller standardized effect size. For the association between ECEC quality and the global measure of child outcomes, 1 standardized unit increase (11.3) in the average age of children was associated with, on average, 0.10 (95% C.I. 0, 0.20) increase in effect size.

*(a) Structural indicators of ECEC quality versus process quality indicators*. We first tested for associations between structural characteristics and child outcomes, and between process quality indicators and child outcomes. In separate analyses, we combined effect sizes for structural characteristics and the effect sizes process quality indicators; effect sizes were pooled with each of the 8 child outcome categories. For structural characteristics, none of the associations were significant (S6 File for the results). For process quality indicators, most child outcome categories showed a significant association. Higher levels of process quality were significantly related to higher levels of academic outcomes (literacy, n = 96: 0.09, 95% C.I. 0.03–0.16; math, n = 56: 0.09, 95% C.I. 0.05–0.12), behavioral skills (n = 64: 0.13, 95% C.I. 0.08–0.18), and social competence (n = 59: 0.14, 95% C.I. 0.08–0.20), and lower levels of behavioral (n = 59: -0.14, 95% C.I. -0.20 - -0.07) and social-emotional problems (n = 27: -0.09, 95% C.I. -0.15 - -0.02). For motor skills (n = 2: 0.09, 95% C.I. -0.02–0.20) and when a global assessment of child outcomes was reported (n = 12: 0.04, 95% C.I. -0.08–0.16), however, the association was not significant.

Table 2 presents the results of the comparison between effect sizes that involved a structural indicator of ECEC quality with those that involved a process quality indicator. The comparison did not consider whether an effect size came from the same study or not. For most child outcomes, the difference in effect size between structural indicators of ECEC quality and process quality indicators was not significant. Two exceptions were found. For math and behavioral skills, the comparison between effect sizes was significant. The results indicated that effect sizes for structural indicators were significantly smaller than for process quality indicators. The findings were robust to the consideration of sex composition and the average age of children in the sample.

Next, we examined whether the effect size for the associations between process quality and child outcomes differed by the three domains of process quality: instructional, emotional (includes the presence of emotional support and absence of conflict), and managerial quality (includes the presence of routines and structures, and absence of chaos). The first set of analyses did not include control variables. In general, associations between the individual domains of process quality and child outcomes were small and mostly non-significant (S7 File). Some

**Table 2. Differences in effect size by ECEC quality indicator.**

| Child Outcome | Included studies (*n*) | Coefficient (SE) | | *t* (df) | | 95% CI lower, upper |
|---|---|---|---|---|---|---|
| Math | 41 | -0.09 | (0.03) | -2.95* | (12.12) | -0.15, -0.02 |
| | 41 | -0.08 | (0.03) | -2.56* | (12.32) | -0.15, -0.01 |
| Language/Literacy | 84 | -0.07 | (0.04) | -1.62 | (25.94) | -0.16, 0.02 |
| | 84 | -0.07 | (0.04) | -1.70 | (26.06) | -0.16, 0.02 |
| Behavioral skills | 55 | -0.08 | (0.03) | -2.77* | (6.31) | -0.15, -0.01 |
| | 55 | -0.08 | (0.03) | -2.57* | (6.41) | -0.15, -0.00 |
| Social competence | 48 | -0.08 | (0.05) | -1.62 | (9.75) | -0.18, 0.03 |
| | 48 | -0.06 | (0.05) | -1.18 | (9.89) | -0.18, 0.05 |
| Behavioral problems | 51 | 0.21 | (0.10) | 2.13 | (10.26) | -0.01, 0.43 |
| | 51 | 0.18 | (0.10) | 1.69 | (10.66) | -0.05, 0.41 |
| Social-emotional problems | 24 | 0.04 | (0.08) | 0.56 | (1.05) | -0.82, 0.90 |
| | 24 | 0.03 | (0.10) | 0.31 | (1.05) | -1.13, 1.19 |
| Global Score | 13 | -0.14 | (0.10) | -1.42 | (3.03) | -0.45, 0.17 |
| | 13 | -0.03 | (0.09) | -0.33 | (3.20) | -0.30, 0.24 |

*Note.* The reference is process quality indicator. The second row reports the analyses that controlled for sex composition (proportion of girls in the sample) and average child age in the sample (in months).

* Difference between coefficients is significant (i.e., 95% CI does not include zero).

significant albeit small associations were found for the instructional quality domain. Higher levels of instructional support by the teacher in the classroom were positively associated with academic outcomes (literacy, n = 81: 0.11, 95% C.I. 0.06–0.15; math, n = 40: 0.11, 95% C.I. 0.07–0.15), behavioral skills (n = 54: 0.12, 95% C.I. 0.06–0.19), and social competence (n = 46: 0.10, 95% C.I. 0.03–0.17), and negatively associated with behavioral (n = 49: -0.12, 95% C.I. -0.20 - -0.04) and social-emotional problems (n = 24: -0.10, 95% C.I. -0.19 - -0.01). When control variables were included, the results were robust and no significant differences between the three domains of process quality were found (S8 File).

*(b) Evidence for moderation.* We tested whether structural indicators of ECEC quality moderated process quality-child outcome associations (S9 File). The results of the moderator analyses were not significant. Only two out of 33 tested moderations were significant. Teacher education moderated the association between process quality and language/literacy, both without (coefficient (SE) = -0.12 (0.04), t(df) = -3.45 (30.11), 95% CI -0.20, -0.00) and with control variables (coefficient (SE) = -0.12 (0.04), t(df) = -2.85 (24.27), 95% CI -0.21, -0.03). Teacher-child ratio moderated the association between process quality and behavioral problems (coefficient (SE) = 0.02 (0.00), t(df) = 7.42 (2.50), 95% CI 0.01, 0.03; The model could only be run without control variables because these were not reported in the available studies.). We further examined whether process quality moderated the associations between structural indicators of ECEC quality and child outcomes (S10 File). The results of the moderator analyses were not significant.

*(c) Method of process quality assessment (observed versus teacher-report).* The next question explored whether different approaches to measuring process quality (observational measures versus self-report measures) gave rise to different or similar associations between process quality indicators and child outcomes. The analysis included a within-study comparison of observational versus self-report measures of process quality. There were eight pairs of effect sizes reported from five studies, which allowed for within-study comparison of process quality-child outcome associations according to the method of process quality measurement. The pairs are represented by each line in Fig 3, comparing the effect sizes based on self-reported

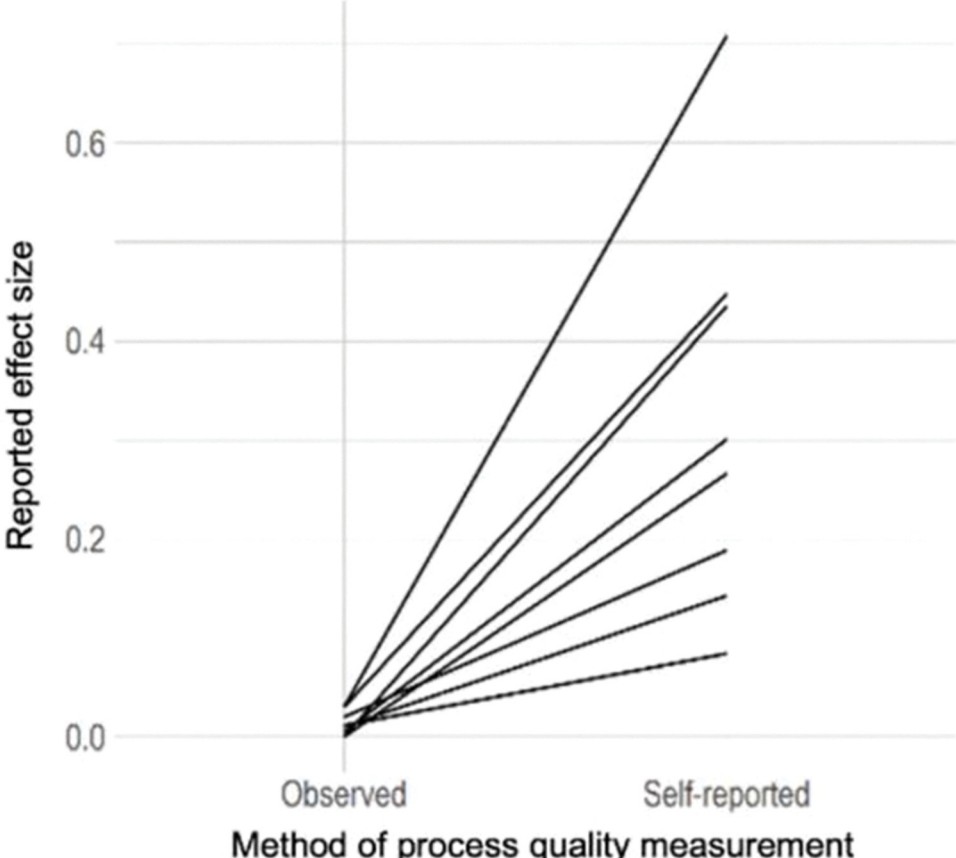

**Fig 3. Comparison of effect sizes based on the method of process quality assessment (self-report versus observation).**

process quality with those based on observed process quality. Results indicated that effect sizes based on self-reported process quality measures were not only higher (mean difference = 0.33, Kruskal-Wallis [KW] chi-squared = 11.33, df = 1, $p$-value = 0.00) but also more variable than the ones based on observed process quality measures.

*(d) Differences by the timing of the data collection (beginning versus end of school year; concurrently versus longitudinally).* This analysis explored whether timing during the ECE year was associated between process quality indicators and child outcomes. There was little evidence to suggest that effect sizes differed depending on whether they were reported in the beginning versus end of the school year (WK chi-squared = 0.71, df = 1, $p$-value = 0.40) or at the same versus different time points during the same school year (KW chi-squared = 0.23, df = 1, $p$-value = 0.63) (S11 File).

*(e) Did ECEC quality-child outcome associations differ by ethnic minority or socioeconomic family background?.* The final question examined the unique associations between ECEC quality indicators and child outcomes when family background differences were taken into account. We used the percentage of children from an ethnic minority and a low-income family background in each study as a moderator of the association between ECEC quality indicators and child outcomes. The results yielded coefficients that were close to zero. The models without control variables detected two significant coefficients. The percentage of children from a low-income family background in a study moderated the association between ECEC quality

indicators and social competence (coefficient (SE) = -0.00 (0.00), t(df) = 2.54 (10.52), 95% CI -0.00, -0.00) and behavioral problems (coefficient (SE) = 0.00 (0.00), t(df) = 2.33 (12.55), 95% CI 0.00, 0.00). However, the coefficients did not remain significant when control variables were included in the models, suggesting that neither the percentage of children from an ethnic minority in a study nor the percentage of children from a low-income family background in a study were associated with the magnitude of effect sizes between ECEC quality indicators and child outcomes (S12 File).

## Discussion

This meta-analysis examined the associations of structural and process indicators of ECEC quality with young children's academic, behavioral and social outcomes in recent studies conducted between 2010 and 2020, a period when widespread acknowledgement of the importance of both types of quality advanced [11] and when quality was incorporated into the wording of global goals for early childhood development [5]. Although the current meta-analysis aimed to synthesized evidence from around the world, the majority of studies reported data from the U.S. (123 out of 185 studies), thus highlighting the need to expand research beyond the U.S. to inform global, regional, and local ECEC quality research, policy-making and practice. Overall, our findings suggest that higher levels of ECEC quality were significantly related to higher levels of academic outcomes, behavioral skills, motor skills, and social competence, and lower levels of behavioral and social-emotional problems, albeit the effect sizes were small. The findings align with previous meta-analyses, suggesting that ECEC quality plays an important role for children's development during the early childhood years [3, 6, 8, 9, 19, 24, 49]. Structural characteristics alone did not significantly relate to child outcomes whereas associations between process quality indicators and most child outcomes were significant, albeit small. A comparison of structural characteristics and process quality indicators, however, did not yield significant differences in effect sizes for most child outcomes. With regard to combined effects of structural and process quality indicators of ECEC on child outcomes, we did not find evidence for moderated associations. We also did not find evidence that ECEC quality-child outcome associations differed by ethnic minority or socioeconomic family background, suggesting that children from various backgrounds benefit from high quality ECEC services.

Overall, the combined effect sizes for structural and process quality indicators of ECEC quality on children's outcomes were small. The present meta-analysis thus complements findings from a recent meta-analysis on the links between ECEC quality and children's outcomes [19] but with a larger sample drawn from all regions of the world. In addition, considerable variation in effect sizes was found, both within and across child outcomes. In our meta-analysis, effect sizes were somewhat larger for behavioral and social-emotional outcomes compared to academic outcomes. It is possible that these findings might hint to different accentuations of ECEC quality effects that depend on the respective child outcome. Supporting children in the development of academic skills may be different than facilitating children's behavioral and social-emotional development [80]. However, differences in effect sizes between child outcomes were small. For this reason, our interpretation remains speculative and further exploration of how ECEC structures and processes differentially influence children's academic, behavioral, and social-emotional skills is warranted.

When tested separately, the results of the present meta-analysis indicate significant, albeit small overall effect sizes for process quality indicators for most child outcomes but not for structural characteristics, suggesting that the combined effect sizes might have been driven by process quality indicators. Numerous studies have demonstrated beneficial effects of high

levels of process quality on child outcomes [13, 17, 20, 22, 26, 31, 55, 65]. It is the immediate experiences of children that arise through interactions and activities, rich in content and stimulation, that are central to children's learning [81]. ECEC policy and practice thus needs to focus on fostering a physical and social environment that enhances positive child developmental outcomes, by providing them with educational material adapted to their needs and accompanied by warm, sensitive interactions [81, 82]. To ensure the consistent implementation of such practices, ECEC standards, often still focused on structural aspects, need to include process quality indicators. Similarly, assessment tools and rating systems used by governments to monitor and evaluate ECEC programs need to place a greater emphasis on process quality [3, 83].

The lack of significant findings for structural characteristics–child outcome associations in the present meta-analysis might be due to the much smaller number of studies for which effect sizes were available. Alternatively, it might have been the selection of structural characteristics included in the meta-analysis that has driven the non-significant findings. The selection was informed by prior research and reflected frequently studied characteristics, such as teacher-child ratio and class size as well as teacher age, education and years of experience. Because efforts to improve ECEC quality in many countries are focused on these characteristics, they received a lot of policy attention that is reflected in regulations and quality monitoring [3, 17]. As a consequence, variation in these structural dimensions might have been limited, resulting in reduced predictive power. In the future, other structural characteristics that are, to date, less frequently studied, such as capacity building of the ECEC workforce and working conditions, need to move into the focus of policy and research to further advance the provision of quality services [7].

When effect sizes of structural characteristics and process quality indicators were compared, differences between the two sets of ECEC quality indicators were largely not significant. Two exceptions were identified. Children seemed to benefit more from process quality indicators compared to structural indicators with regard to early math development and behavioral skills. Promoting early math skills requires very specific stimulation and, as such, high-quality processes may be more critical to math development than structural indicators of ECEC quality [19]. Similarly, process quality has been suggested to be the key educational driver supporting the development of behavioral skills critical for learning, such as self-regulation and positive learning behaviors [84]. It is important to point out though that the comparison did not consider whether an effect size came from the same study or not. As such, the results could be driven by differences between studies rather than differences between the two sets of ECEC quality indicators. The next step will be to identify differential effects of ECEC quality indicators on children's outcomes in order to best support optimal learning.

In addition to investigating direct associations between ECEC quality indicators and child outcomes, we also tested moderation. Overall, we did not find consistent evidence that structural indicators of ECEC quality moderated process quality-child outcome associations nor did we find consistent evidence that process quality moderated the associations between structural indicators of ECEC quality and child outcomes. These findings are in contrast with theoretical assumptions that it is the combination of structural and process indicators of ECEC quality that affects children's development [6, 81]. However, the lack of findings in the present meta-analysis aligns with the scarce empirical support for this theoretical assumption. If moderated effects were found in prior research, they were only found for few structural-process indicator combinations and were smaller than direct effects (for a review, [85]. The lack of stronger results may be due to the fact that structural indicators of ECEC quality explain little variance in process quality indicators [85]. This may be particularly true in countries with strong national regulation and monitoring of structural quality, such as the U.S. and many

Western European countries, where there is limited variation in structural indicators. It is also important to note that only a few studies included both structural and process indicators of ECEC quality and even fewer studies focused on the same child outcome. Thus, we might not have had enough power to detect moderation effects.

Overall, associations between ECEC quality and child outcomes were not influenced by the percentage of ethnic-minority children in a study nor by the percentage of children from a low-income family background in a study. The findings complement recent meta-analytic findings, suggesting that the beneficial effects of ECEC quality on children's outcomes can be found for children from various family backgrounds [19]. Together, the meta-analytic evidence supports the notion that high-quality learning can be one strategy to ensure that children are prepared for school [44]. However, it will be important to "sustain the boost that quality preschool education can provide" beyond ECEC [46, p. 31]. The effects of ECEC on child outcomes are more likely sustained when children transition to higher quality schools [46]. However, children from disadvantaged family backgrounds are more likely to attend high-poverty, segregated schools and have less educated school teachers [44]. What is needed are research-based specific practices and policies that address the root causes of educational disparities. For that, research must consider "children's development in the context of the child-care system as well as the family system, and recognize the links between these systems and the larger society" [86, p. 165].

In addition, we found that self-report measures of process quality indicators yielded higher and more variable effect sizes compared to observational measures. These findings might have been influenced by biases common to self-report measures, i.e., self-representation bias or social desirability bias [37]. For example, it is possible that teachers overestimated the quality of interactions with children in their classroom. Similar results, i.e., teachers overestimating the quality of their instructional practices, have been found in previous research and were explained by an individual's desire to avoid their responses reflecting negatively on them or contradicting common values and expectations of their group [37]. Observational measures have the potential to counter these issues; however, their implementation in research and practice is time- and cost-intensive. Although observational measures are "preferable for seizing the learning potential of ECE centers [...] the development of more economic but equally reliable and valid alternatives is necessary" [19, p. 1484].

The final results of the present meta-analysis related to the time in the school year when ECEC quality and child outcome measures were collected. We did not find evidence to suggest that effect sizes differed depending on whether they were reported in the beginning versus end of the school year or at the same versus different time points during the same school year.

## Limitations and future directions

Despite the attempt to provide a synthesis of the global literature on ECEC quality and child outcome associations, the vast majority of studies eligible for the present meta-analyses included samples from the U.S. In addition, studies with large samples (>500 participants) were also predominantly from the U.S. Together, the results might have been biased towards patterns prevalent in the U.S. that might not apply to other, non-U.S. ECEC contexts. It is possible that the use of English-language databases and the English requirement for studies to be included in the coding have resulted in studies from non-English majority speaking countries being underrepresented in the data. Increased efforts and resources are needed to overcome the challenges of locating, assessing and including non-English studies in systematic reviews, for example, by using professional translators [87]. Another limitation related to the dominance of studies from the U.S. might be related to the measures used to assess ECEC quality.

For example, 81 studies, of which 50 were from the U.S., used a version of the CLASS, an observational tool developed in the U.S. to assess indicators of process quality. As a result, other measures, such as the ECERS-R and ECERS-E were not as commonly reported which might have biased the results towards a certain conceptualization of ECEC quality. Relatedly, conceptual and theoretical frameworks that have been derived from research conducted in the U.S. might not apply to ECEC classrooms in other countries [88]. ECEC services in many countries have undergone significant changes over the past years [19]. To build knowledge about children's educational opportunities across the world, establishing a global research agenda on ECEC will be critical, with a particular emphasis on the role of ECEC on children's outcomes [19]. The present meta-analysis only included short-term outcomes, restricted to the same school year. An additional limitation was that many studies did not include both, structural and process quality indicators in the analyses, and/or multiple child outcomes. Thus, only a limited number of studies was available for testing the mechanisms underlying the associations between ECEC quality indicators and child outcomes. For this reason, we grouped the available effect sizes by child outcome and thus could not test all possible interactions of ECEC quality indicators. Similarly, not all studies reported detailed demographic (age, sex, ethnic minority status, family SES) information about the children participating in the study. Finally, the present meta-analysis relied on correlational effects. Zero-order correlations do not reflect the complexity of ECEC classrooms, pointing to the need of innovative meta-analytic approaches that allow for the aggregation of published multivariate findings [19].

## Conclusion

The results of the present meta-analysis suggest that ECEC quality indicators affect a broad range of developmental outcomes. The implementation of high-quality ECEC services needs to be guided by an extended list of ECEC quality standards that go beyond traditional classroom- and teacher-level structural characteristics and include additional, less frequently studied structural aspects, such as capacity building of the ECEC workforce and working conditions, as well as process quality indicators. The enhancement of ECEC quality standards also needs to be reflected in assessment tools and rating systems used by governments to monitor and evaluate ECEC programs. If informed by scientific evidence, such a shift in ECEC quality standards can help maximize the benefits of ECEC participation. In addition, more rigorous research is needed to better understand the unique and combined effects of multiple ECEC quality dimensions (and child and family characteristics) to identify effective ways for quality improvement and meeting children's unique needs. For that, a more nuanced analysis of structural and process quality, and of child and family characteristics, is needed to understand the multiple dimensions to consider for quality improvement.

## Supporting information

**S1 Checklist. PRISMA 2020 for abstracts checklist.**
(DOCX)

**S2 Checklist. PRISMA 2020 checklist.**
(DOCX)

**S1 File. List of keywords for literature search.**
(DOCX)

**S2 File. Descriptive information of studies included in the meta-analysis.**
(XLSX)

**S3 File. Funnel plots.**
(DOCX)

**S4 File. Sensitivity analysis.**
(DOCX)

**S5 File. Association between ECEC quality and child outcomes (only studies from high-income countries).**
(DOCX)

**S6 File. Associations between structural characteristics and child outcomes.**
(DOCX)

**S7 File. Differences in ECEC quality–Child outcome associations by the type of process quality domain.**
(DOCX)

**S8 File. Differences in effect size by process quality domain.**
(DOCX)

**S9 File. Tests of structural indicators of ECEC quality as moderators of process quality-child outcome associations.**
(DOCX)

**S10 File. Tests of process quality indicators of ECEC quality as moderators of structural quality-child outcome associations.**
(DOCX)

**S11 File. Differences by the timing of the data collection.**
(DOCX)

**S12 File. Differences by ethnic minority or socioeconomic family background.**
(DOCX)

**S13 File.**
(ZIP)

## Author Contributions

**Conceptualization:** Antje von Suchodoletz, Hirokazu Yoshikawa.

**Data curation:** Antje von Suchodoletz, D. Susie Lee, Junita Henry, Supriya Tamang, Bharathy Premachandra.

**Formal analysis:** D. Susie Lee.

**Investigation:** Antje von Suchodoletz.

**Methodology:** Antje von Suchodoletz, Junita Henry, Supriya Tamang, Bharathy Premachandra, Hirokazu Yoshikawa.

**Project administration:** Antje von Suchodoletz.

**Resources:** Antje von Suchodoletz.

**Supervision:** Antje von Suchodoletz, Bharathy Premachandra.

**Visualization:** Antje von Suchodoletz, D. Susie Lee, Junita Henry, Supriya Tamang.

**Writing – original draft:** Antje von Suchodoletz.

**Writing – review & editing:** Antje von Suchodoletz, Junita Henry, Hirokazu Yoshikawa.

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
