## [Decision Letter · Decision Letter 0]

16 Jan 2023

PONE-D-22-31943Early Childhood Education and Care Quality and Associations with Child Outcomes: A Meta-AnalysisPLOS ONE

Dear Dr. Von Suchodolotz,

Thank you for submitting your manuscript to PLOS ONE. After careful consideration, we feel that it has merit but does not fully meet PLOS ONE’s publication criteria as it currently stands. Therefore, we invite you to submit a revised version of the manuscript that addresses the points raised during the review process.

We look forward to receiving your revised manuscript.

Kind regards,

Sze Yan Liu, PhD

Academic Editor

PLOS ONE

Journal Requirements:

3. Please ensure that you include a title page within your main document. You should list all authors and all affiliations as per our author instructions and clearly indicate the corresponding author.

Additional Editor Comments:

This is a well-written study. I agree with the reviewers that while the analysis is generally clear the text could benefit from more details.

Reviewers' comments:

Reviewer's Responses to Questions

**Comments to the Author**

1. Is the manuscript technically sound, and do the data support the conclusions?

Reviewer #1: Yes

Reviewer #2: Yes

2. Has the statistical analysis been performed appropriately and rigorously? 

Reviewer #1: Yes

Reviewer #2: Yes

3. Have the authors made all data underlying the findings in their manuscript fully available?

Reviewer #1: Yes

Reviewer #2: Yes

4. Is the manuscript presented in an intelligible fashion and written in standard English?

Reviewer #1: Yes

Reviewer #2: Yes

5. Review Comments to the Author

Reviewer #1: PONE-D-22-31943 Review

This is a thoughtful and well-written meta-analysis examining the influence of ECEC program structural and process indicators of quality on child outcomes. It is clear that the authors put a tremendous amount of work into this review, which is an important contribution to the literature. I have listed some major and several minor comments below.

Abstract:

Can the structural indicators of ECEC quality and process quality indicators that were associated with child outcomes be described in the abstract? It remains vague to just state that indicators of quality (in general) are associated with various child outcomes.

Introduction

Line 105-109: It would be useful for authors to expand on why it is important to continue to examine whether structural indicators of ECEC quality are associated with child outcomes, when they have not been shown to matter in previous work. Why is further investigation needed to inform policy and practice.

Methods

Line 241: Why did authors exclude studies prior to 2010? The rationale in the article was that a previous meta-analysis (Rao published 2017) had gone up to 2012. But that analysis did not attempt to answer the questions about quality, but rather different types of programs such as child-focused or parent-directed or nutrition. A number of pre-2010 papers could be included, for example:

Aboud, F. E. (2006). Evaluation of an early childhood preschool program in rural Bangladesh. Early Childhood Research Quarterly, 21, 46–60. doi:10.1016/j.ecresq.2006.01.008

Moore, A. C., Akhter, S., & Aboud, F. E. (2008). Evaluating an improved quality preschool program in rural Bangladesh. International Journal of Educational Development, 28, 118–131. doi:10.1016/j.ijedudev.2007.05.003.

Mwaura, P., Sylva, K., & Malmberg, L.-E. (2008). Evaluating the Madrasa pre-school programme in East Africa: A quasi experimental study. International Journal of Early Years Education, 16, 237–255.

The supporting excel sheet listing studies and their measures appears to exclude research using the ECERS-E as the measure of quality. Correlations with the ECERS-E tend to be higher than the ECERS-R and studies using the measure have been frequently conducted in LMICs and in Britain.

It is not clear why longitudinal studies, when a child outcome came from a time after the quality measure, and intervention studies were excluded. Why would their associations be irrelevant to the questions asked here? These two features are most likely to exclude LMIC studies where interventions are often the only ethical reason for conducting such a study.

It appears that 6 interventions were excluded. The number of longitudinal studies excluded is not reported.

Several publications after 2010 were omitted. It would be important to include these especially as they are from LMIC, which the authors claim to be lacking:

Malmberg L-E, Mwaura P, Sylva K. Effects of a preschool intervention on cognitive development among East-African preschool children: A flexibly time-coded growth model. Early Child Res Q 2011;26(1):124-33.

Aboud, Frances E., Kerrie Proulx, and Zaitu Asrilla. An impact evaluation of Plan Indonesia’s early childhood program. Canadian Journal of Public Health 107.4 (2016): e366-e372.

Su, Yufen, et al. Preschool quality and child development in China. Early childhood research quarterly 56 (2021): 15-26.

Aboud, F.E. & Hossain, K (2011). The impact of preprimary school on primary school achievement in Bangladesh. Early Childhood Research Quarterly, 26, 237-246.

PLOS recent published a meta-analysis of parenting programs, separating out high-income country findings from LMICs. Could the same be done here? Out of 185 studies listed in the excel sheet, 165 were from HICs. This is not representative of the quality-outcome research conducted in LMICs. Perhaps you can conduct one analysis for HIC and a separate one for LMIC studies, adding more LMIC studies than currently (see comments above).

Line 289: please specify for the readers what is meant by a “global process quality score”?

Please specify how each estimate of association is weighted when calculating the pooled effect size.

Line 295. The five structural qualities were clear. However, the four process qualities were not. How did you categorize CLASS and ECERS-R items into these four process qualities?

Results:

Two questions were posed: "whether such structural characteristics itself systematically change the effects of process quality on child outcomes, or whether process quality changes the impact of structural characteristics on child outcomes. Could you also ask and present the results for the two simpler questions before the moderated ones, namely: Do structural characteristics impact child outcomes and Do process characteristics impact child outcomes?

Where are the individual measures of association in each study presented? Meta-analyses typically present the data extracted from each study that contributes to the analyses.

Figure 1: I would expect that effect sizes would differ depending on the indicator of quality (i.e., type of structural and type of process indicators of quality). Why were these not separated, and effect sizes for child outcomes calculated for each indicator?

Figure 1: why does the size of the circle not represent the number of effect sizes (rather than unique studies) used to estimate the pooled effect size? It seems it should be number of estimates of association since some studies had multiple estimates of association. Also, in the results section (e.g., paragraph starting on line 484), does the n represent number of studies or number of effect sizes used to estimate the pooled effect size?

Table 2: please make clear which type of quality indicator is used as the reference (I believe it is process). In the text, you state that effect sizes for associations that include process indicators are more positive than those that include structural indicators, yet the regression coefficient estimates in the table are negative. This is confusing. I suggest authors stay consistent in the way they discuss and present the direction of associations.

Figure SI 5: While this figure is nice, it would benefit from also listing the estimates and 95% CIs for the pooled effect sizes.

Table S2: I see that instructional quality was used as the reference group. But there are two other groups, so why do we not see how each group differs from the reference?

(b) Evidence for moderation: Where are the non-significant results presented?

Line 484. Can you comment on whether the effect sizes were small, moderate or large? They all appear to be small and Literacy and math appear to be very small.

Line 589: The authors state that there was significant moderation from family income on the association between quality indicators and social competence and behavioral problems. However, the coefficients are 0 (95% CI: 0-0). Please explain.

In moderation analyses, it is typical to see effect sizes for each stratum (e.g., high vs low proportion of children from low-income families). What do the coefficients in Figure S5 represent? Is this the coefficient for the interaction term? If so, please make this explicit in the Figure. If not, please explain and clarify what the coefficient represents.

Discussion

Line 699. It is difficult to draw conclusions about frameworks and evidence from LMICs unless you add more research from LMICs and conduct analyses comparing HICs and LMICs.

Line 712. You stated that the reliance on correlation coefficients is a limitation. What kind of analysis would be more appropriate?

Reviewer #2: I appreciate the opportunity to review the meta-analysis on early childhood education and care quality and child outcomes. The study is well done with clear rationales and descriptions of the methods and results. I believe the findings will add to the literature on ECEC quality and children’s development. I provide specific comments below but want to emphasize that I think the authors need to be clear that the effect sizes found are small and more information is needed on the practical significance of the findings. Additionally, the implication section is underdeveloped, and more effort should be put into discussing how these findings fit with the broader literature and what this means for practice and policy.

Literature review is well written and thorough, with the exception of the discussion of the interaction of quality indicators. The justification for interaction effects is underdeveloped – why would one think that an association between structural aspects and child outcomes will be stronger with higher levels of process quality? The theoretical model of structure – process – outcome would not predict this. Unclear what is motivating this question. Also, do the authors have any hypotheses for which quality elements together are most predictive of child outcomes or how the combination may differ depending on outcome examined?

Section on Children from Ethnic Minority Backgrounds appropriately and importantly highlights the challenges students may encounter and how their background can contribute to differences in achievement. However, evidence from Head Start and other pre-K evaluations suggest that multilingual learners may benefit the most from the ECE (see work by Marianne Bitler and other on Head Start, NC preK RCT evaluation results).

Given the focus on structural and process quality, I was surprised the authors did not discuss policy more in the introduction and literature review as how quality in programs is regulated. This is particularly important in the global context where policies differ widely and may contribute to differences observed/reported in studies.

Methods – overall this section is well done. I have just a few specific comments below.

• Does the requirement of English result in studies from non-English majority speaking countries not being represented in the data?

• What is the rationale for dropping intervention studies?

• Did the authors attempt to contact study authors to obtain any missing information?

• After looking at the excel database, I was surprised to find some key ECE studies not there (Soliday Hong et al., 2019; Keys et al., 2013; Burchinal et al., 2016). I am guessing this is because of the studies including meta-analyses of the datasets. I think a further description of the methods used to achieve the final sample, even in the supplementary files, would help readers better understand decisions.

Results are clearly written and nicely organized.

Discussion section

• Overall, the section is thorough. However, the authors should emphasize throughout, particularly in the first paragraph that the overall effect is small for all child outcomes. This is done nicely in the second paragraph.

• I’m not sure what to make of the significant differences between structural and process quality indicators for the two outcomes – the authors should expand on these findings. Such as is this a data issue or is there reason to believe the findings are meaningful and aligned with prior research.

• I’m unsure what is meant by two sentences on page 32-33 “While, at the sample level, these results seem to be robust…when studying the nature of the effects of ECE quality.”

• The authors should situate their effect sizes in the literature and provide information on the practical significance. Also, understanding the cost of improving quality is important and is not equal for all indicators. More should be said on this point.

• The conclusion section is very broad and generally mentions quality improvement. As discussed above, this can look quite different depending on which areas of quality are the focus (e.g., structural – requiring teachers to have certain degrees vs. improving process quality). Right now, I think the implications are too general and not particularly useful to the field.

6. PLOS authors have the option to publish the peer review history of their article (what does this mean?). If published, this will include your full peer review and any attached files.

Reviewer #1: No

Reviewer #2: No

---

## [Author Response · Author response to Decision Letter 0]

27 Mar 2023

15-03-2023

We wish to thank the reviewers for their valuable comments. In the revised manuscript, we addressed all comments. The detailed responses are listed below and in the response letter.

Reviewer #1: PONE-D-22-31943 Review

This is a thoughtful and well-written meta-analysis examining the influence of ECEC program structural and process indicators of quality on child outcomes. It is clear that the authors put a tremendous amount of work into this review, which is an important contribution to the literature. I have listed some major and several minor comments below.

Abstract:

Can the structural indicators of ECEC quality and process quality indicators that were associated with child outcomes be described in the abstract? It remains vague to just state that indicators of quality (in general) are associated with various child outcomes.

RESPONSE: Many thanks for your comments. In the analyses, indicators of ECEC quality were combined to maximize the number of included effect sizes. As such, the results do not differentiate between indicators. We are now more explicit about this in the abstract by adding: The averaged effects, pooled within each of the child outcomes, suggest that higher levels of ECEC quality are significantly related child outcomes (line 24). 

In response to another comment raised by the reviewer, we now added separate analyses for structural and process quality indicators. The results from these analyses are now also reported in the abstract (lines 31-35): When structural and process quality indicators were tested separately, structural characteristics alone did not significantly relate to child outcomes whereas associations between process quality indicators and most child outcomes were significant, albeit small. A comparison of structural characteristics and process quality indicators, however, did not yield significant differences in effect sizes for most child outcomes.

Introduction

Line 105-109: It would be useful for authors to expand on why it is important to continue to examine whether structural indicators of ECEC quality are associated with child outcomes, when they have not been shown to matter in previous work. Why is further investigation needed to inform policy and practice.

RESPONSE: In response to the comment, we expanded on why it is important to examine structural indicators of ECEC quality. In lines 114-122, we added:

Because structural features of ECEC settings have been more regulable, many countries focus on structural standards as key strategy for improving the quality of ECEC programs (Early, et al., 2007; OECD, 2018). The G20 Development Working Group (2018) urges to focus on the quality of the infrastructure and capacity building of, decent work conditions for and adequate training of the ECEC workforce. Government regulations can set standards for these features, for example, by lifting the minimum requirements for teacher-child ratio or requiring a certain percentage of teaching staff to be qualified in early childhood education. Such structural regulations determine the setting in which children learn and thus may be important preconditions for process quality (OECD, 2018).

Methods

Line 241: Why did authors exclude studies prior to 2010? The rationale in the article was that a previous meta-analysis (Rao published 2017) had gone up to 2012. But that analysis did not attempt to answer the questions about quality, but rather different types of programs such as child-focused or parent-directed or nutrition. A number of pre-2010 papers could be included, for example:

Aboud, F. E. (2006). Evaluation of an early childhood preschool program in rural Bangladesh. Early Childhood Research Quarterly, 21, 46–60. doi:10.1016/j.ecresq.2006.01.008

Moore, A. C., Akhter, S., & Aboud, F. E. (2008). Evaluating an improved quality preschool program in rural Bangladesh. International Journal of Educational Development, 28, 118–131. doi:10.1016/j.ijedudev.2007.05.003.

Mwaura, P., Sylva, K., & Malmberg, L.-E. (2008). Evaluating the Madrasa pre-school programme in East Africa: A quasi experimental study. International Journal of Early Years Education, 16, 237–255.

RESPONSE: Many thanks for pointing this out. We have now focused on previous meta-analyses that attempt to answer questions related to ECEC quality and child outcomes. A number of meta-analyses included articles published prior to 2010 which is why we decided to keep 2010 as year to start our literature search. We added this information in the revised manuscript in lines 274-278: 

We based these dates on previous meta-analyses investigating associations between various aspects of ECEC quality and child outcomes for which literature searches included articles published prior to 2010 (e.g., Brunsek, et al., 2017; Burchinal et al., 2011; Eggert et al., 2018; Falenchuk et al., 2017; Hong, et al., 2019; Jensen & Rasmussen, 2019; Perlman, et al., 2016; Perlman, et al., 2017).

We also added the following (lines 280-285):

The time period (2010-2020) also covers a period of ECEC-focused policy initiatives across the world, for example, the national plan for medium and long-term education reform and development (2010-2020) in China (Zhou, 2015), recommendations on high-quality ECEC systems by the Council of the European Union (2019), or the G20 Initiative for Early Childhood Development (2018) and G20 Education Ministers’ Declaration (2018).

The supporting excel sheet listing studies and their measures appears to exclude research using the ECERS-E as the measure of quality. Correlations with the ECERS-E tend to be higher than the ECERS-R and studies using the measure have been frequently conducted in LMICs and in Britain.

RESPONSE: This is an important point. The inclusion of studies in the meta-analysis was not based on the measures used to assess indicators of ECEC quality. Rather, the inclusion was based on whether the study reported an effect size relevant to the research questions (i.e., an association between ECEC quality and child outcomes). The lack of research using the ECERS-E as a measure of ECEC quality was not a result of the inclusion/exclusion criteria used in the present meta-analysis. 

It is also possible that many studies using the ECERS as a measure of ECEC quality were not included because the studies were published prior to 2010. For example, in a recent meta-analysis testing the relationship between the Early Childhood Environment Rating Scale and its revised form and child outcomes, 68% of included studies (51 out of 75 as per supplemental information S3) were published prior to 2010.

Brunsek, A., Perlman, M., Falenchuk, O., McMullen, E., Fletcher, B., & Shah, P. S. (2017). The relationship between the Early Childhood Environment Rating Scale and its revised form and child outcomes: A systematic review and meta-analysis. PloS one, 12(6), e0178512.

In lines 809-815, we addressed your comment in the limitation:

Another limitation related to the dominance of studies from the U.S. might be related to the measures used to assess ECEC quality. For example, 81 studies, of which 50 were from the U.S., used a version of the CLASS, an observational tool developed in the U.S. to assess indicators of process quality. As a result, other measures, such as the ECERS-R and ECERS-E were not as commonly reported which might have biased the results towards a certain conceptualization of ECEC quality.

It is not clear why longitudinal studies, when a child outcome came from a time after the quality measure, and intervention studies were excluded. Why would their associations be irrelevant to the questions asked here? These two features are most likely to exclude LMIC studies where interventions are often the only ethical reason for conducting such a study.

RESPONSE: Thank you for this comment. We apologize for not being clear in the description of inclusion criteria. The comment relates to one eligibility criterium (line 298): “the study assessed indicators of quality in center-based ECEC programs catering to children ages 0-6 years” and two inclusion criteria (reported in lines 303-304): “the article reported effect size measure of at least one quality indicator-child outcome association” and “measures of ECEC quality and child outcomes were collected within the same school year”. A longitudinal study was included if an effect size measure of at least one ECEC quality indicator-child outcome association was available when both ECEC quality and child outcomes were assessed during the same school year of the ECEC program.

There were two reasons for why longitudinal studies were excluded:

(1) when child outcome measures were from a different school year than ECEC quality measures. To avoid bias introduced to the results, longitudinal studies with child outcomes measures from a different school year than the ECEC quality measure were excluded. There was no consistency in the length of the time when ECEC quality was measured and child outcomes were measured across longitudinal studies. The length ranged between 1-10 years which would need to be analyzed separately by length of follow-up and there were not enough effect sizes to permit such an analysis.

(2) when child outcome measures were assessed before ECEC quality measures. In this case, ECEC quality was no longer the predictor of child outcomes but rather predicted by child outcomes which was not our research question.

It appears that 6 interventions were excluded. The number of longitudinal studies excluded is not reported.

Several publications after 2010 were omitted. It would be important to include these especially as they are from LMIC, which the authors claim to be lacking:

Malmberg L-E, Mwaura P, Sylva K. Effects of a preschool intervention on cognitive development among East-African preschool children: A flexibly time-coded growth model. Early Child Res Q 2011;26(1):124-33.

Aboud, Frances E., Kerrie Proulx, and Zaitu Asrilla. An impact evaluation of Plan Indonesia’s early childhood program. Canadian Journal of Public Health 107.4 (2016): e366-e372.

Su, Yufen, et al. Preschool quality and child development in China. Early childhood research quarterly 56 (2021): 15-26.

Aboud, F.E. & Hossain, K (2011). The impact of preprimary school on primary school achievement in Bangladesh. Early Childhood Research Quarterly, 26, 237-246.

RESPONSE: The comment relates to an exclusion criterium: “Intervention studies were excluded unless relevant effect size measures were reported prior to the intervention.” (reported in line 308). The decision was made to avoid bias to the results that might have been due to the intervention. In an attempt to standardize the conditions in which effect sizes were reported as much as possible across studies, only pre-intervention (baseline) effect sizes were included from intervention studies. That was because interventions targeted ECEC quality and as such aimed to change (increase) ECEC quality, thus reflecting very different conditions from studies that observed ECEC quality but did not attempt to increase ECEC quality. The 6 excluded intervention studies did not report such effect sizes and were therefore excluded.

PLOS recent published a meta-analysis of parenting programs, separating out high-income country findings from LMICs. Could the same be done here? Out of 185 studies listed in the excel sheet, 165 were from HICs. This is not representative of the quality-outcome research conducted in LMICs. Perhaps you can conduct one analysis for HIC and a separate one for LMIC studies, adding more LMIC studies than currently (see comments above).

RESPONSE: Thank you for the suggestion. As the reviewer mentioned, the number of studies from LMICs that could be identified by our literature search during the a-priori defined search period was small and did not allow to run separate analysis. In response to the comment, however, we did run a separate analysis for studies from HIC countries only for the research question regarding the association between ECEC quality and child outcomes. The results are reported in the supplemental materials (SI5).

We added the following to the revised manuscript (lines 549-552):

We repeated the analysis using only studies from high-income countries. The results replicated and are reported Supplemental Information SI 5. However, for low-to-middle income countries, the same separate analysis could not be completed because of the small number of unique studies.

Line 289: please specify for the readers what is meant by a “global process quality score”?

RESPONSE: More details are now provided in the revised manuscript (lines 343-345).

If the study used a global process-quality score (i.e., the score did not differentiate between specific indicators but instead reflected an average level of process quality across multiple indicators) this was recorded as a separate process quality indicator.

Please specify how each estimate of association is weighted when calculating the pooled effect size.

RESPONSE: The information has been added to the revised manuscript (lines 401-404).

We used robust variance estimation to calculate the pooled effect sizes in which weights of each estimate of association were based on the assumption that effect sizes are correlated within studies. Details on the formulas specifying the correlated effects covariance structure and weights calculation can be found in Fisher and Tipton (2015, p. 4).

Line 295. The five structural qualities were clear. However, the four process qualities were not. How did you categorize CLASS and ECERS-R items into these four process qualities?

RESPONSE: In the revised manuscript we now provide additional information for how items or scales were categorized (lines 341-343).

To categorize items or scales of measures into these four indicators, we relied on the description of the measure and labeling by the author(s) of the original study.

Results:

Two questions were posed: "whether such structural characteristics itself systematically change the effects of process quality on child outcomes, or whether process quality changes the impact of structural characteristics on child outcomes. Could you also ask and present the results for the two simpler questions before the moderated ones, namely: Do structural characteristics impact child outcomes and Do process characteristics impact child outcomes?

RESPONSE: Thank you for the suggestion. We now added the recommended separate analyses and tested for associations between structural characteristics and child outcomes, and between process quality indicators and child outcomes.

The results are reported in the main manuscript (lines 579-591): 

We first tested for associations between structural characteristics and child outcomes, and between process quality indicators and child outcomes. In separate analyses, we combined effect sizes for structural characteristics and the effect sizes process quality indicators; effect sizes were pooled with each of the 8 child outcome categories. For structural characteristics, none of the associations were significant (see Supplement Information SI6 for the results). For process quality indicators, most child outcome categories showed a significant association. Higher levels of process quality were significantly related to higher levels of academic outcomes (literacy, n=96: 0.09, 95% C.I. 0.03 – 0.16; math, n=56: 0.09, 95% C.I. 0.05 – 0.12), behavioral skills (n=64: 0.13, 95% C.I. 0.08 – 0.18), and social competence (n=59: 0.14, 95% C.I. 0.08 – 0.20), and lower levels of behavioral (n=59: -0.14, 95% C.I. -0.20 - -0.07) and social-emotional problems (n=27: -0.09, 95% C.I. -0.15 - -0.02). For motor skills (n=2: 0.09, 95% C.I. -0.02 – 0.20) and when a global assessment of child outcomes was reported (n=12: 0.04, 95% C.I. -0.08 – 0.16), however, the association was not significant.

And in Supplement Information (SI 6):

SI6: Associations between structural characteristics and child outcomes

The associations between structural characteristics and child outcomes were not significant (literacy, n=28: 0.03, 95% C.I. -0.02 – 0.08; math, n=16: 0.01, 95% C.I. -0.04 – 0.05; behavioral skills, n=9: 0.01, 95% C.I. -0.04 – 0.05; social competence, n=13: 0.03, 95% C.I. -0.03 – 0.08; behavioral problems, n=13: -0.03, 95% C.I. -0.07 - 0.13; social-emotional problems, n=2: -0.02, 95% C.I. -0.88 - 0.84; motor skills, n=2: 0.14, 95% C.I. -0.61 – 0.89; and global assessment of child outcomes, n=5: -0.05, 95% C.I. -0.26 – 0.15).

Where are the individual measures of association in each study presented? Meta-analyses typically present the data extracted from each study that contributes to the analyses.

RESPONSE: Thank you for your comment. The documents are available in the Data and Results supplemental materials:

Excel document (Coding Sheet): (FINAL) Coding Sheet_11_25_22.xlsx

Excel document (Effect Sizes): ECEC Quality_A Meta-Analysis_all_effect_sizes.xlsx

Figure 1: I would expect that effect sizes would differ depending on the indicator of quality (i.e., type of structural and type of process indicators of quality). Why were these not separated, and effect sizes for child outcomes calculated for each indicator?

RESPONSE: Thank you for the comment. To examine the overall magnitude of associations between ECEC quality indicators and child outcomes, we averaged effects pooled within each of the child outcomes. Structural and process indicators of ECEC quality were combined to maximize the number of included effect sizes. Figure 1 presents the pooled effect size estimates for ECEC quality-child outcome associations. 

In response to a previous comment, we now added the recommended separate analyses and tested for associations between structural characteristics and child outcomes, and between process quality indicators and child outcomes. For the analysis testing associations between structural characteristics and child outcomes, we combined all structural indicators of ECEC quality. Similarly, for the analysis testing associations between process quality indicators and child outcomes, we combined all process quality indicators of ECEC quality. This was done to maximize the number of included studies. Because of the large confidence intervals observed for the separate analyses, we decided to no present the findings in Figure 1. However, the results are presented in the main manuscript (lines 579-591) and the Supplement Information (SI 6).

Figure 1: why does the size of the circle not represent the number of effect sizes (rather than unique studies) used to estimate the pooled effect size? It seems it should be number of estimates of association since some studies had multiple estimates of association. Also, in the results section (e.g., paragraph starting on line 484), does the n represent number of studies or number of effect sizes used to estimate the pooled effect size?

RESPONSE: Thank you for the comment. We used the Robust MA method that takes into account that some studies contributed multiple effect sizes. Thus, the number of effect sizes might be misleading as they could come from a small number of studies. For this reason, we decided to report the number of unique studies.

In the revised manuscript, we added to the results section, that n represents the number of unique studies (line 539).

Table 2: please make clear which type of quality indicator is used as the reference (I believe it is process). In the text, you state that effect sizes for associations that include process indicators are more positive than those that include structural indicators, yet the regression coefficient estimates in the table are negative. This is confusing. I suggest authors stay consistent in the way they discuss and present the direction of associations.

RESPONSE: Thank you for the comment. As per the reviewer’s suggestion, we added the information to the table note. The reference is process quality. We also revised the text to align with the table.

Figure SI 5: While this figure is nice, it would benefit from also listing the estimates and 95% CIs for the pooled effect sizes.

RESPONSE: The requested information has been added to the figure.

Table S2: I see that instructional quality was used as the reference group. But there are two other groups, so why do we not see how each group differs from the reference?

RESPONSE: In Table S2, we now report three coefficients for each child outcome. The statistics reported in the first row for each outcome reflect managerial quality (in reference to instructional quality); the statistics reported in the second row for each outcome reflect emotional quality (in reference to instructional quality); the statistics reported in the second row for each outcome reflect emotional quality (in reference to managerial quality). We included this information in the table note.

(b) Evidence for moderation: Where are the non-significant results presented?

RESPONSE: The results are available in the Data and Results supplemental materials; Excel document: 220513_results_meta-analyses_Revision1_10032023.xlsx (Sheets (b) moderation struct and (b) moderation process. We now also added the results to Supplement Information Table S3 (Tests of structural indicators of ECEC quality as moderators of process quality-child outcome associations) and Table S4 (Tests of process quality indicators of ECEC quality as moderators of structural quality-child outcome associations).

Line 484. Can you comment on whether the effect sizes were small, moderate or large? They all appear to be small and Literacy and math appear to be very small.

RESPONSE: This has been added (line 539).

Line 589: The authors state that there was significant moderation from family income on the association between quality indicators and social competence and behavioral problems. However, the coefficients are 0 (95% CI: 0-0). Please explain.

RESPONSE: Thank you for the comment. In the manuscript, we only reported two decimals. Because the numbers were very small, it rounded to 0.00. The actual numbers are as follows: 

Social competence: coefficient = -0.0028, SE = 0.0011, 95% Confidence Interval = -0.0053, -0.0003

Behavioral problems: coefficient = 0.0029, SE = 0.0012, 95% Confidence Interval = -0.0002, 0.0056

The detailed results are available in the Data and Results supplemental materials; Excel document: 220513_results_meta-analyses_Revision1_10032023.xlsx (Sheet (e) minority SES).

In moderation analyses, it is typical to see effect sizes for each stratum (e.g., high vs low proportion of children from low-income families). What do the coefficients in Figure S5 represent? Is this the coefficient for the interaction term? If so, please make this explicit in the Figure. If not, please explain and clarify what the coefficient represents.

RESPONSE: Thank you for the comment. The coefficients reported in the figure can be interpreted as the extent to which the association between ECEC quality and child outcomes changes if the percentage of children from an ethnic minority or a low-income family background increase by 10%. We added this information to the figure caption/note.

Discussion

Line 699. It is difficult to draw conclusions about frameworks and evidence from LMICs unless you add more research from LMICs and conduct analyses comparing HICs and LMICs.

RESPONSE: Many thanks for the comment. The discussion about conceptual and theoretical frameworks is not specifically focused on a comparison between HICs and LMICs. We agree with the reviewer, that we do not have the data to make such a claim. Rather, we wanted to point out that the majority of identified studies that were included in the meta-analysis included samples from the U.S. (123 studies compared to 62 studies from countries other than the U.S.). In addition, studies with large samples (>500 participants) were also predominantly from the U.S. We concluded that the results might have been biased towards patterns prevalent in the U.S. that might not apply to other, non-U.S. ECEC contexts.

Line 712. You stated that the reliance on correlation coefficients is a limitation. What kind of analysis would be more appropriate?

RESPONSE: In response to the comment, we added the following to the revised manuscript (lines: 830-831):

Zero-order correlations do not reflect the complexity of ECEC classrooms, pointing to the need of innovative meta-analytic approaches that allow for the aggregation of published multivariate findings (Ulferts et al., 2019). 

Reviewer #2: I appreciate the opportunity to review the meta-analysis on early childhood education and care quality and child outcomes. The study is well done with clear rationales and descriptions of the methods and results. I believe the findings will add to the literature on ECEC quality and children’s development. I provide specific comments below but want to emphasize that I think the authors need to be clear that the effect sizes found are small and more information is needed on the practical significance of the findings. Additionally, the implication section is underdeveloped, and more effort should be put into discussing how these findings fit with the broader literature and what this means for practice and policy.

Literature review is well written and thorough, with the exception of the discussion of the interaction of quality indicators. The justification for interaction effects is underdeveloped – why would one think that an association between structural aspects and child outcomes will be stronger with higher levels of process quality? The theoretical model of structure – process – outcome would not predict this. Unclear what is motivating this question. Also, do the authors have any hypotheses for which quality elements together are most predictive of child outcomes or how the combination may differ depending on outcome examined?

RESPONSE: Many thanks for the comment. In response to the feedback, we added more information to the discussion of the interaction of ECEC quality indicators. This specific aspect of the meta-analysis was largely exploratory which is why we did not have specific hypotheses.

The following was added to the introduction (lines 181-204):

A better understanding of the underlying processes linking ECEC quality with child outcomes may be gained by testing interaction effects of ECEC quality indicators. It is possible that it is a specific combination of structural and process aspects that matters for children’s outcomes. For example, it has been found that associations between process quality and children’s social-emotional skills were moderated by dosage. Children who spent more time in high-quality ECEC settings were reported to have higher levels of social-emotional skills compared to children who spent less time in high-quality ECEC settings (Aguiar, 2016; Cunha et al., 2006). Such results suggest that structural characteristics can reinforce positive effects of high levels of process quality as well as negative effects of low levels of process quality. Likewise, positive effects of the level of instructional processes on children’s gains in literacy and numeracy might only be present in small classes where teachers can engage in differentiated instruction, whereas in large classes such an effect might be absent. However, results are mixed and other studies did not find significant results when testing structural characteristics of the ECEC setting as moderators of the association between process quality and child outcomes (Xue et al., 2016).

Alternatively, it might also be possible that associations between structural aspects and child outcomes will be stronger under high levels of process quality, compared to low levels of process quality. For example, a study found that teacher emotional support moderated the association between classroom composition (i.e., high levels of problem behaviors in the classroom) and children’s relational functioning. The negative effect of a highly challenging class on individual children’s relational functioning was buffered by teachers who were highly emotionally supportive (Buyse et al., 2008). Although fewer studies tested the moderating role of process quality, it can provide important information about the mechanisms underlying the associations between ECEC quality indicators and child outcomes.

Section on Children from Ethnic Minority Backgrounds appropriately and importantly highlights the challenges students may encounter and how their background can contribute to differences in achievement. However, evidence from Head Start and other pre-K evaluations suggest that multilingual learners may benefit the most from the ECE (see work by Marianne Bitler and other on Head Start, NC preK RCT evaluation results).

RESPONSE: In response to the reviewer’s comment, we elaborated the discussion and included a note on ECEC program evaluations (lines 208-218).

Indeed, evaluations of ECEC programs (for example, Head Start in the U.S.) provide evidence for this assumption, suggesting that program effects may be largest for children from disadvantaged backgrounds (Bitler et al., 2014; Iruka, 2020; Weiland & Yoshikawa, 2013; Yoshikawa et al., 2016). ECEC programs have the potential to compensate for educational disadvantages by providing rich and engaging learning environments and to support these children to catch up with their peers (Ulfers & Anders, 2015). As such, ECEC programs can disrupt trends leading to achievement gaps which have been found to start prior to age three (Iruka, 2020). Yet, to date, systems, including ECEC, continue to perpetuate racism and inequities, thus “reduc[ing] opportunities for certain groups to thrive and meet their potential” (Iruka, 2020, p.65). In order to strengthen the impact of early learning, more effective, evidence-based policies are thus needed.

Given the focus on structural and process quality, I was surprised the authors did not discuss policy more in the introduction and literature review as how quality in programs is regulated. This is particularly important in the global context where policies differ widely and may contribute to differences observed/reported in studies.

RESPONSE: Thank you for the important comment. In the revised introduction, we now refer to policy where we saw fit, with a focus on global policy initiatives.

We added the following to the manuscript:

Moreover, the G20 Initiative for Early Childhood Development (2018) emphasizes the importance of political buy-in and state and non-state investments in the early years in order to narrow achievement and opportunity gaps that exist between children from higher and lower socioeconomic backgrounds. (lines 58-61)

Because structural features of ECEC settings have been more regulable, many countries focus on structural standards as key strategy for improving the quality of ECEC programs (Early, et al., 2007; OECD, 2018). The 2018 G20 Development Working Group (2018) urges to focus on the quality of the infrastructure and capacity building of, decent work conditions for and adequate training of the ECEC workforce. Government regulations can set standards for these features, for example, by lifting the minimum requirements for teacher-child ratio or requiring a certain percentage of teaching staff to be qualified in early childhood education. Such structural regulations determine the setting in which children learn and are thus important preconditions for process quality (OECD, 2018). (lines 114-122)

Methods – overall this section is well done. I have just a few specific comments below.

• Does the requirement of English result in studies from non-English majority speaking countries not being represented in the data?

RESPONSE: At the pre-screening stage, 168 studies were excluded because they were either in another language (non-English) or not available online. This reflects 4% of studies excluded at this stage. Six additional studies were identified at the screening stage that were also not in English and thus excluded which reflects 1% of excluded studies during the screening stage. See the PRISMA flowchart of article selection for this information. As such, the exclusion of studies because they were not published in English was a minor reason. In addition, the exclusion of non-English publications is a common exclusion criteria for meta-analysis (see, for example, Cosso, Suchodoletz, & Yoshikawa, 2022; Eggert, Fukkink, & Eckhardt, 2018; Perlman, et al., 2016; Perlman et al., 2017).

However, it is possible that our search did not pick up relevant studies from non-English majority speaking countries as they might have been published in journals that are not listed with the databases used. We added this as a possible limitation (lines 804-809):

It is possible that the use of English-language databases and the English requirement for studies to be included in the coding have resulted in studies from non-English majority speaking countries being underrepresented in the data. Increased efforts and resources are needed to overcome the challenges of locating, assessing and including non-English studies in systematic reviews, for example, by using professional translators (Neimann Rasmussen & Montgomery, 2018).

• What is the rationale for dropping intervention studies?

RESPONSE: The comment relates to an exclusion criterium: “Intervention studies were excluded unless relevant effect size measures were reported prior to the intervention.” (reported in line 308). The decision was made to avoid bias to the results that might have been due to the intervention. In an attempt to standardize the conditions in which effect sizes were reported as much as possible across studies, only pre-intervention (baseline) effect sizes were included from intervention studies. That was because interventions targeted ECEC quality and as such aimed to change (increase) ECEC quality, thus reflecting very different conditions from studies that observed ECEC quality but did not attempt to increase ECEC quality. The 6 excluded intervention studies did not report such effect sizes and were therefore excluded.

• Did the authors attempt to contact study authors to obtain any missing information?

RESPONSE: Yes, we contacted authors to obtain missing information, yet, the response rate was only 32%. 

• After looking at the excel database, I was surprised to find some key ECE studies not there (Soliday Hong et al., 2019; Keys et al., 2013; Burchinal et al., 2016). I am guessing this is because of the studies including meta-analyses of the datasets. I think a further description of the methods used to achieve the final sample, even in the supplementary files, would help readers better understand decisions.

RESPONSE: Meta-analyses, literature reviews or systematic syntheses were excluded. The information is reported in the PRISMA flowchart. We now added a sentence to the method section that meta-analyses, literature reviews and systematic syntheses of multiple datasets were excluded during the screening phase (line 306).

Results are clearly written and nicely organized.

RESPONSE: Thank you for the positive feedback.

Discussion section

• Overall, the section is thorough. However, the authors should emphasize throughout, particularly in the first paragraph that the overall effect is small for all child outcomes. This is done nicely in the second paragraph.

RESPONSE: In the first paragraph, we now added that effect sizes were small (line 682).

• I’m not sure what to make of the significant differences between structural and process quality indicators for the two outcomes – the authors should expand on these findings. Such as is this a data issue or is there reason to believe the findings are meaningful and aligned with prior research.

• The authors should situate their effect sizes in the literature and provide information on the practical significance. Also, understanding the cost of improving quality is important and is not equal for all indicators. More should be said on this point.

• The conclusion section is very broad and generally mentions quality improvement. As discussed above, this can look quite different depending on which areas of quality are the focus (e.g., structural – requiring teachers to have certain degrees vs. improving process quality). Right now, I think the implications are too general and not particularly useful to the field.

RESPONSE: In response to the reviewer’s comments, we have thoroughly revised the discussion and conclusion and incorporated the reviewer’s suggestions (lines 671-846).

• I’m unsure what is meant by two sentences on page 32-33 “While, at the sample level, these results seem to be robust…when studying the nature of the effects of ECE quality.”

RESPONSE: The sentences were deleted during the revision.

---

## [Decision Letter · Decision Letter 1]

7 May 2023

Early Childhood Education and Care Quality and Associations with Child Outcomes: A Meta-Analysis

PONE-D-22-31943R1

Dear Dr. Von Suchodolotz,

We’re pleased to inform you that your manuscript has been judged scientifically suitable for publication and will be formally accepted for publication once it meets all outstanding technical requirements.

Kind regards,

Sze Yan Liu, PhD

Academic Editor

PLOS ONE

Additional Editor Comments (optional):

Thank you for the revisions and the clarifications.

Reviewers' comments:

Reviewer's Responses to Questions

**Comments to the Author**

1. If the authors have adequately addressed your comments raised in a previous round of review and you feel that this manuscript is now acceptable for publication, you may indicate that here to bypass the “Comments to the Author” section, enter your conflict of interest statement in the “Confidential to Editor” section, and submit your "Accept" recommendation.

Reviewer #1: All comments have been addressed

2. Is the manuscript technically sound, and do the data support the conclusions?

Reviewer #1: Yes

3. Has the statistical analysis been performed appropriately and rigorously? 

Reviewer #1: Yes

4. Have the authors made all data underlying the findings in their manuscript fully available?

Reviewer #1: Yes

5. Is the manuscript presented in an intelligible fashion and written in standard English?

Reviewer #1: Yes

6. Review Comments to the Author

Reviewer #1: The authors have thoroughly and thoughtfully addressing my previous comments. I have no further suggestions.

7. PLOS authors have the option to publish the peer review history of their article (what does this mean?). If published, this will include your full peer review and any attached files.

Reviewer #1: No

---

## [Editor Report · Acceptance letter]

12 May 2023

PONE-D-22-31943R1 

Early Childhood Education and Care Quality and Associations with Child Outcomes: A Meta-Analysis 

Dear Dr. von Suchodoletz:

I'm pleased to inform you that your manuscript has been deemed suitable for publication in PLOS ONE. Congratulations! Your manuscript is now with our production department. 

Kind regards, 

on behalf of

Dr. Sze Yan Liu 

Academic Editor

PLOS ONE